# HOLDOUT-LOSS-BASED DATA SELECTION FOR LLM FINETUNING VIA IN-CONTEXT LEARNING

**Ling Zhang**[*] **& Xianliang Yang**[*]
Microsoft Research Asia
Beijing, China
{zhangling, xianya}@microsoft.com

**Juwon Yu & Park Cheonyoung**
Korean KT
Seoul, Korea
{juwon1.yu, cheonyoung}@kt.com

**Lei Song & Jiang Bian**
Microsoft Research Asia
Beijing, China
{Lei.Song, Jiang.Bian}@microsoft.com

## ABSTRACT

Fine-tuning large pretrained language models is a common approach for aligning them with human preferences, but noisy or off-target examples can dilute supervision. While small, well-chosen datasets often match the performance of much larger ones, systematic and efficient ways to identify high-value training data remain underexplored. Many current methods rely on heuristics or expensive retraining. We present a principled, resource-efficient framework for data selection and reweighting. At its core is an In-Context Approximation (ICA) that estimates the holdout loss a model would incur after training on a candidate example by conditioning on a small, curated holdout set in context. ICA requires no reference model and no additional finetuning. We define the resulting estimate as the ICA score, and derive per-example weights that dynamically reweight gradient updates as model parameters evolve. Across SFT, DPO, and SimPO, and over diverse backbones and datasets, ICA-based reweighting consistently improves model alignment with minimal overhead [1]. We analyze sensitivity to score update frequency and the number of in-context holdout examples. We also discuss limitations in rapidly drifting on-policy settings, highlighting directions for future work.

## 1 INTRODUCTION

Fine-tuning has become the standard approach for adapting large pretrained language models to downstream applications and aligning them with human intent (Wei et al., 2021; Ouyang et al., 2022). This process typically starts with supervised fine-tuning (SFT) on instruction-response pairs (Wei et al., 2021), followed by preference-based alignment using pairwise preference data, e.g., RLHF (Christiano et al., 2017; Ouyang et al., 2022), DPO (Rafailov et al., 2023), and simPO (Meng et al., 2024). Since fine-tuning effectively steers the pretrained model toward desired behaviors, the quality of training data plays a central role: high-quality examples provide clear alignment signals, while noisy or inconsistent data can severely degrade performance.

However, training data for fine-tuning, typically collected from human annotators or model-generated outputs, often contain errors, inconsistencies, or redundancies. For example, Gao et al. (2024) report that 20–40% of preference pairs in LM alignment are noisy, and that alignment performance is highly sensitive to such noise. In addition, recent studies have shown that smaller but carefully curated datasets can yield alignment performance comparable to much larger ones (Zhou et al., 2023; Chen et al., 2023). Despite the recognized importance and promise of effective data selection for fine-tuning, systematic yet principled approaches remain underexplored.

---

[*]Equal contribution.

[1]Code and prompts are available at `https://github.com/microsoft/HeurAgenix/tree/dpo`.

A central challenge is the absence of consensus on what constitutes "valuable" data. While some approaches adopt hand-crafted heuristics or leverage large models as judges, a more principled perspective is to define data value by its impact on downstream performance, which is the ultimate goal of fine-tuning. Nevertheless, directly measuring each example's contribution to downstream performance would require retraining (and evaluating) the model across all possible candidates, a process that is computationally infeasible.

To overcome this challenge, prior work has explored different strategies, including influence-function-based methods that approximate model performance change using Taylor expansions (Pruthi et al., 2020; Xia et al., 2024; Wang et al., 2024), surrogate models that fit a linear approximation of the loss (Engstrom et al., 2024), and bilevel optimization or meta-learning approaches (Shen et al., 2024; Grangier et al., 2023; Calian et al., 2025). Beyond these analytical and learning-based approaches, empirical studies examine correlations between hand-picked notions of data quality and downstream performance to inform heuristic data selection criteria (Liu et al., 2023; Bukharin & Zhao, 2023; Zhao et al., 2024; Lu et al., 2023; Li et al., 2023a; Cao et al., 2023; Li et al., 2023b). These methods, while promising, are often computationally expensive or lack theoretical grounding.

In this work, we propose a principled, resource-efficient framework for data selection in model fine-tuning. Following prior work (Mindermann et al., 2022; Xia et al., 2024; Calian et al., 2025), we aim to select a subset of training data such that the model trained on it minimizes the holdout loss, i.e., the loss on the holdout set. Each example's value is quantified based on the holdout loss the model would incur if trained with that example. Computing this naively is intractable, so we build on a framework introduced by RHO-Loss (Mindermann et al., 2022), which uses Bayesian probability theory to derive a more tractable expression for the holdout loss. Nonetheless, efficiently computing this expression remains challenging, as evaluating it would normally require retraining at each step.

RHO-Loss mitigates per-step retraining by relying on a separate reference model trained on the holdout set. We introduce an in-context approximation (ICA) that eliminates the need for both additional fine-tuning and a fixed reference model. Motivated by the insight from (Dai et al., 2023), we provide the holdout set as in-context demonstrations at each training step to approximate the effects of explicit parameter updates on model performance. The resulting data values derived from ICA, termed ICA scores, enable dynamic evaluation of each example's utility as the model evolves. These scores are then used to reweight gradient updates during fine-tuning, prioritizing examples that most reduce holdout loss. Experiments show that training with ICA-based reweighting consistently improves model alignment for SFT, DPO, and SimPO across diverse datasets and backbone models.

## 2 RELATED WORK

A central question in data selection is how to determine what makes a training example valuable. This is often done by quantifying each example's impact on a downstream proxy, typically measured as the loss on a small, high-quality holdout set. Methods in this category include influence-function formulations (Pruthi et al., 2020; Xia et al., 2024; Wang et al., 2024), Data Shapley (Ghorbani & Zou, 2019; Wang et al., 2025), and learned scorers such as Datamodels (Engstrom et al., 2024), meta-learning frameworks (Calian et al., 2025), and optimization-based approaches (Grangier et al., 2023; Shen et al., 2024; Gu et al., 2025; Pan et al., 2025). Particularly relevant to our work is RHO-Loss (Mindermann et al., 2022), which estimates the holdout loss a model would incur if trained on a particular example, but requires a separate reference model. Another related approach is One-Shot Learning (Li et al., 2023b), which also leverages in-context learning, though in a different manner from our method.

In addition to analytic or learned approaches for quantifying data value, hand-crafted notions of data quality have been studied, with empirical analyses assessing how these proxies correlate with fine-tuned model performance and inform data selection (Liu et al., 2023; Bukharin & Zhao, 2023; Zhao et al., 2024; Lu et al., 2023; Li et al., 2023a; Cao et al., 2023; Huang & Goyal, 2025; Yu et al., 2025; Deng et al., 2025; Morimura et al., 2024). Beyond quantifying data value with respect to downstream performance, prior work has also explored selecting data to directly match the target distribution, using techniques such as gradient alignment (Killamsetty et al., 2021), importance resampling (Xie et al., 2023; Katharopoulos & Fleuret, 2018), or optimal transport (Kang et al., 2024). In contrast to these data-centric methods, a separate line adopts an algorithmic perspective, directly modify-

ing learning objectives and proposing extensions to SFT or DPO to account for instance-specific differences or improve generalization (Wu et al., 2024; D'Oosterlinck et al., 2025; Wu et al.; 2025).

## 3    PRELIMINARY

**In-context learning**    LLMs have demonstrated strong in-context learning (ICL) capabilities (Brown et al., 2020; Dong et al., 2022). In ICL, a few demonstration examples are concatenated with a query to form the model input. The model then identifies patterns from these examples and makes predictions without any parameter updates.

Formally, given a query $\mathbf{x}$, the model predicts an output $\mathbf{y}$ conditioned on a demonstration set $C$. In this work, we adopt a demonstration set of the following form, which includes an optional task instruction $I$ followed by $k$ demonstration examples:

$$C = \{I, s(\mathbf{x}_1, \mathbf{y}_1), \ldots, s(\mathbf{x}_k, \mathbf{y}_k)\},$$

where $s(\cdot, \cdot)$ is a formatting function (Dong et al., 2022).

**Supervised fine-tuning**    SFT adapts pretrained LLMs to produce responses with desired characteristics using instruction–response pairs. Let $\mathcal{D} = \{(\mathbf{x}_i, \mathbf{y}_i^\star)\}_{i=1}^{|\mathcal{D}|}$ denote the instruction dataset, where $\mathbf{y}_i^\star$ is the reference response for query $\mathbf{x}_i$. SFT updates the model parameters by minimizing the sentence-level cross-entropy:

$$\mathcal{L}_{\text{SFT}}(\boldsymbol{\theta}) = \mathbb{E}_{(\mathbf{x}, \mathbf{y}^\star) \sim \mathcal{D}}\big[ -\log \pi_{\boldsymbol{\theta}}(\mathbf{y}^\star \mid \mathbf{x})\big]$$

where $\pi_{\boldsymbol{\theta}}(\mathbf{y} \mid \mathbf{x})$ is the probability of generating $\mathbf{y}$ given $\mathbf{x}$ under the model parameterized by $\boldsymbol{\theta}$.

**Preference-based alignment methods**    RLHF (Christiano et al., 2017; Ouyang et al., 2022) is a widely used method to align LLMs with human preferences. It uses data of the form $\mathcal{D} = \{(\mathbf{x}_i, \mathbf{y}_{w,i}, \mathbf{y}_{l,i})\}_{i=1}^{|\mathcal{D}|}$, where $\mathbf{y}_{w,i}$ and $\mathbf{y}_{l,i}$ are the preferred and dispreferred responses for a prompt $\mathbf{x}_i$. The standard RLHF pipeline first learns a reward model and then optimizes a policy using RL algorithms such as PPO.

DPO (Rafailov et al., 2023) provides an off-policy alternative that bypasses the RL step, learning directly from preference data without training a reward model. Specifically, DPO solves:

$$\mathcal{L}_{\text{DPO}}(\boldsymbol{\theta}) = -\mathbb{E}_{(\mathbf{x}, \mathbf{y}_w, \mathbf{y}_l) \sim \mathcal{D}}\Big[ \log \sigma \Big(\beta \frac{\pi_{\boldsymbol{\theta}}(\mathbf{y}_w \mid \mathbf{x})}{\pi_{\text{ref}}(\mathbf{y}_w \mid \mathbf{x})} - \beta \frac{\pi_{\boldsymbol{\theta}}(\mathbf{y}_l \mid \mathbf{x})}{\pi_{\text{ref}}(\mathbf{y}_l \mid \mathbf{x})}\Big)\Big]$$

where $\pi_{\text{ref}}$ is a reference policy, $\sigma$ is the sigmoid function, and $\beta$ controls the trade-off between adhering to the reference and incorporating new preference data.

A recent variant, SimPO (Meng et al., 2024), extends DPO by eliminating the need for a reference model and introducing a length-normalized implicit reward based on the average log probability of a sequence. Its objective is:

$$\mathcal{L}_{\text{simPO}}(\boldsymbol{\theta}) = -\mathbb{E}_{(\mathbf{x}, \mathbf{y}_w, \mathbf{y}_l) \sim \mathcal{D}}\Big[ \log \sigma \Big(\frac{\beta}{|\mathbf{y}_w|} \pi_{\boldsymbol{\theta}}(\mathbf{y}_w \mid \mathbf{x}) - \frac{\beta}{|\mathbf{y}_l|} \pi_{\boldsymbol{\theta}}(\mathbf{y}_l \mid \mathbf{x}) - \gamma\Big)\Big]$$

where $\gamma > 0$ is a target reward margin ensuring that the reward difference between winning and losing responses exceeds this threshold.

## 4    HOLDOUT-LOSS-BASED DATA SELECTION VIA IN-CONTEXT LEARNING

In this section, we formally introduce the problem formulation and present a holdout-loss-based data selection framework that leverages in-context learning for efficient computation.

### 4.1    PROBLEM FORMULATION

We consider the problem of fine-tuning a pretrained model on a large, but potentially noisy, training dataset $\mathcal{D} = \{(\mathbf{x}_i, \mathbf{y}_i)\}_{i=1}^{|\mathcal{D}|}$, where $(\mathbf{x}, \mathbf{y})$ may represent an instruction–response pair or a preference

pair. The goal is to optimize model performance on a smaller, higher-quality holdout set $\mathcal{D}_{\text{ho}} = \{(\mathbf{x}_i^{\text{ho}}, \mathbf{y}_i^{\text{ho}})\}_{i=1}^{|\mathcal{D}_{\text{ho}}|}$. For example, $\mathcal{D}_{\text{ho}}$ could be a corrected subset of $\mathcal{D}$ generated by a stronger model or a manually curated set from a target domain.

However, training on all of $\mathcal{D}$ may be inefficient and suboptimal due to the presence of noise. While one could train directly on $\mathcal{D}_{\text{ho}}$, its small size often leads to overfitting and fails to leverage the information in the larger set. Therefore, we aim to select a subset $\bar{\mathcal{D}}^\star \subset \mathcal{D}$ such that a model trained on $\bar{\mathcal{D}}^\star$ minimizes the loss on $\mathcal{D}_{\text{ho}}$ (i.e., the holdout loss). Formally, the problem can be framed as

$$\bar{\mathcal{D}}^\star = \arg \min_{\bar{\mathcal{D}} \subset \mathcal{D}} \mathcal{L}(\mathcal{D}_{\text{ho}}; \boldsymbol{\theta}^\star(\bar{\mathcal{D}})), \quad \text{where} \quad \boldsymbol{\theta}^\star(\bar{\mathcal{D}}) = \arg \min_{\boldsymbol{\theta}} \mathcal{L}(\bar{\mathcal{D}}; \boldsymbol{\theta}). \tag{1}$$

We denote by $\boldsymbol{\theta}^\star(\mathcal{S})$ the model parameters obtained by training on any dataset $\mathcal{S}$ and $\mathcal{L}(\mathcal{S}; \boldsymbol{\theta})$ the loss on $\mathcal{S}$ under model $\boldsymbol{\theta}$.

Equation 1 would be prohibitively expensive to solve naively, as it requires training on every candidate subset $\bar{\mathcal{D}} \subset \mathcal{D}$ and evaluating the holdout loss. Instead of solving it directly, we recast the problem in terms of quantifying the contribution of each training example to reducing holdout loss. Concretely, we assign a score to each example that reflects this contribution. These scores can then be used either to select the example that most reduces the holdout loss, or to reweight gradient updates in the optimizer according to each example's contribution.

In the following section, we describe an in-context learning–based approach for approximating these scores, removing the need for auxiliary retraining.

## 4.2 Computing Approximated Holdout Loss via In-Context Learning

**Bayesian formulation of the holdout loss** Consider sequential (greedy) data selection, where points are added one at a time. Let $\mathcal{D}_t$ be the dataset selected up to step $t$, and $\boldsymbol{\theta}_t := \boldsymbol{\theta}^\star(\mathcal{D}_t)$ the model trained on it. Then, the subset selection problem in Equation 1 reduces to selecting, at each step, the example $(\mathbf{x}, \mathbf{y}) \in \mathcal{D}$ that, when added to $\mathcal{D}_t$, minimizes the holdout loss:

$$(\mathbf{x}^\star, \mathbf{y}^\star) = \arg \min_{(\mathbf{x}, \mathbf{y}) \in \mathcal{D}} \mathcal{L}(\mathcal{D}_{\text{ho}}; \boldsymbol{\theta}^\star(\mathcal{D}_t \cup \{(\mathbf{x}, \mathbf{y})\})). \tag{2}$$

Accordingly, each training example's contribution can be measured by this holdout loss.

To avoid explicit retraining when evaluating the holdout loss in equation 2, we build on the Bayesian formulation introduced by Mindermann et al. (2022). Specifically, considering the negative log-likelihood as the loss function ($\ell(\mathbf{y} \mid \mathbf{x}; \boldsymbol{\theta}) = -\log p(\mathbf{y} \mid \mathbf{x}; \boldsymbol{\theta})$), and, applying Bayes' rule under the conditional independence assumption, the holdout loss of the model trained with a particular example can be expressed as (see Appendix A for a reproduction of the derivation in Mindermann et al. (2022)) [2]:

$$\mathcal{L}(\mathcal{D}_{\text{ho}}; \boldsymbol{\theta}^\star(\mathcal{D}_t \cup \{(\mathbf{x}, \mathbf{y})\})) = \ell(\mathbf{y} \mid \mathbf{x}; \boldsymbol{\theta}^\star(\mathcal{D}_t \cup \mathcal{D}_{\text{ho}})) - \ell(\mathbf{y} \mid \mathbf{x}; \boldsymbol{\theta}_t) - \mathcal{L}(\mathcal{D}_{\text{ho}}; \boldsymbol{\theta}_t) \tag{3}$$

where we use $\mathcal{L}$ for losses over a set, and $\ell$ for the per-example loss. Omitting the term independent of $(\mathbf{x}, \mathbf{y})$ and reversing the sign, we define the remaining expression as the holdout-loss score of each example at step $t$:

$$s_{\text{ho}}(\mathbf{x}, \mathbf{y}; \boldsymbol{\theta}_t) = \ell(\mathbf{y} \mid \mathbf{x}; \boldsymbol{\theta}_t) - \ell(\mathbf{y} \mid \mathbf{x}; \boldsymbol{\theta}^\star(\mathcal{D}_t \cup \mathcal{D}_{\text{ho}})). \tag{4}$$

The optimal example can then be approximately found by maximizing this score:

$$(\mathbf{x}^\star, \mathbf{y}^\star) = \arg \max_{(\mathbf{x}, \mathbf{y}) \in \mathcal{D}} s_{\text{ho}}(\mathbf{x}, \mathbf{y}; \boldsymbol{\theta}_t). \tag{5}$$

The holdout-loss score provides a tractable tool for data selection. However, since $\mathcal{D}_t$ is updated with each newly added example, the model trained on $\mathcal{D}_t \cup \mathcal{D}_{\text{ho}}$ must also be updated at every selection step, which still incurs substantial computational overhead. To mitigate this cost, prior work (Mindermann et al., 2022) approximates the retraining by training a model only on $\mathcal{D}_{\text{ho}}$ and

---

[2]This Bayesian framework also extends to pairwise preference data (e.g., for DPO and simPO), where $\mathbf{y}$ represents a preference pair $(\mathbf{y}_w, \mathbf{y}_l)$ with $\mathbf{y}_w \succ \mathbf{y}_l$, and the loss is defined correspondingly for the chosen preference learning model (e.g., Bradley-Terry as used in DPO).

reusing it across all selection steps, i.e., $\ell(\mathbf{y} \mid \mathbf{x}; \boldsymbol{\theta}^{\star}(\mathcal{D}_t \cup \mathcal{D}_{\mathrm{ho}})) \approx \ell(\mathbf{y} \mid \mathbf{x}; \boldsymbol{\theta}^{\star}(\mathcal{D}_{\mathrm{ho}}))$. This results in the reducible holdout loss (RHO-Loss) criterion employed for data selection in their work.

Nonetheless, RHO-Loss still incurs the cost of training a separate reference model. Moreover, using a fixed model in place of one that would be updated at each selection step can introduce bias in estimating each example's contribution (Wang et al., 2024). To enable an efficient and adaptive data selection criterion, we introduce an in-context approximation that removes auxiliary training entirely by approximating $\ell(\mathbf{y} \mid \mathbf{x}; \boldsymbol{\theta}^{\star}(\mathcal{D}_t \cup \mathcal{D}_{\mathrm{ho}}))$ via in-context learning. This technique is described in detail below.

**Efficient computation via in-context learning**  Motivated by the finding that in-context learning can induce model behaviors similar to those obtained through gradient-based updates (Dai et al., 2023), we provide the holdout set as in-context demonstrations to a model trained on $\mathcal{D}_t$ to approximate the effect of training on the combined set $\mathcal{D}_t \cup \mathcal{D}_{\mathrm{ho}}$. Specifically, the second term in equation 4 is approximated as

$$\ell(\mathbf{y} \mid \mathbf{x}; \boldsymbol{\theta}^{\star}(\mathcal{D}_t \cup \mathcal{D}_{\mathrm{ho}})) \approx \ell(\mathbf{y} \mid \mathbf{x}, \mathcal{D}_{\mathrm{ho}}; \boldsymbol{\theta}_t). \tag{6}$$

We call the approximation in equation 6 the in-context approximation (ICA).

Applying this approximation to equation 4 yields the in-context approximation score (ICA score):

$$s_{\mathrm{ICA}}(\mathbf{x}, \mathbf{y}; \boldsymbol{\theta}_t) := \ell(\mathbf{y} \mid \mathbf{x}; \boldsymbol{\theta}_t) - \ell(\mathbf{y} \mid \mathbf{x}, \mathcal{D}_{\mathrm{ho}}; \boldsymbol{\theta}_t). \tag{7}$$

We can now use the ICA score to approximately identify the data point that maximizes the holdout-loss score in equation 5. This approach is not only more computationally efficient, but also enables dynamic re-evaluation of each example's impact on the holdout loss as the model evolves.

In practice, examples are often selected in batches rather than sequentially. We next describe how to leverage the ICA score for batch selection using a reweighting strategy.

### 4.3 BATCH SELECTION VIA A REWEIGHTING STRATEGY

**ICA score-based reweighting**  For batch selection, we use a reweighting strategy that up-weights high-scoring examples and downweights lower-scoring ones. Unlike hard selection, which only chooses top examples and may reduce batch diversity or ignore interactions among data points (Wang et al., 2024), reweighting leverages the gradient signal from the entire batch, improving training stability. We apply ICA score-based reweighting in our main experiments and ablate this choice in Section 5.3, comparing it to percentile-based filtering.

Concretely, consider gradient-based training (e.g., Adam or SGD) with mini-batch updates. At each iteration $t$, a batch $B_t \subset \mathcal{D}$ of size $n_B$ is sampled. For each example in the batch, we compute its ICA score as defined in equation 7 and convert these scores into continuous weights in the range $[0, 1]$ via max-min normalization. We adopt max–min instead of the softmax used by Wang et al. (2024); Calian et al. (2025) because it preserves the relative differences between scores and avoids the exponential amplification that can distort the contribution of low- and high-scoring examples. The per-example weights are given by

$$w(\mathbf{x}_i, \mathbf{y}_i; \boldsymbol{\theta}_t) = \frac{s(\mathbf{x}_i, \mathbf{y}_i; \boldsymbol{\theta}_t) - \min_{j \in B_t} s(\mathbf{x}_j, \mathbf{y}_j; \boldsymbol{\theta}_t)}{\max_{j \in B_t} s(\mathbf{x}_j, \mathbf{y}_j; \boldsymbol{\theta}_t) - \min_{j \in B_t} s(\mathbf{x}_j, \mathbf{y}_j; \boldsymbol{\theta}_t)}. \tag{8}$$

These weights, reflecting the relative utility of each example for minimizing the holdout loss, are used to scale their contributions to the batch gradient:

$$\mathbf{g}_t = \sum_{i=1}^{|B_t|} w(\mathbf{x}_i, \mathbf{y}_i; \boldsymbol{\theta}_t) \, \nabla_{\boldsymbol{\theta}} \ell(\mathbf{x}_i, \mathbf{y}_i; \boldsymbol{\theta}_t). \tag{9}$$

The resulting weighted gradient $\mathbf{g}_t$ is used to update the model parameters via a standard optimizer.

Because the ICA score, and hence the weights, evolve throughout training, this reweighting strategy adaptively adjusts each example's contribution to the gradient updates according to the holdout loss the model would achieve if trained on that example. Algorithm 1 outlines how the ICA score is computed and used to reweight gradient updates during fine-tuning.

---

**Algorithm 1** Reweighting training examples using ICA scores for LLM fine-tuning

---

1: **Input:** Training set $\mathcal{D}$; Holdout set $\mathcal{D}_{\text{ho}}$; Pre-trained model parameters $\boldsymbol{\theta}$; Number of training steps $T$; Batch size $n_B$; Optimizer OPTIMIZER

2: Initialize $\boldsymbol{\theta}_0 \leftarrow \boldsymbol{\theta}$
3: **for** $t = 0, \cdots, T - 1$ **do**
4:     Sample candidate set $B_t \subset \mathcal{D}$ of size $n_B$
5:     **for** $i = 1, \cdots, n_B$ **do**
6:         $\texttt{ConditionalLoss}_i \leftarrow \ell(\mathbf{y}_i \mid \mathbf{x}_i, \mathcal{D}_{\text{ho}}; \boldsymbol{\theta}_t)$
7:         $\texttt{Loss}_i \leftarrow \ell(\mathbf{y}_i \mid \mathbf{x}_i; \boldsymbol{\theta}_t)$
8:         $\texttt{Score}_i \leftarrow \texttt{Loss}_i - \texttt{ConditionalLoss}_i$
9:     **end for**
10:     Compute per-example weights within the batch using equation 8
11:     Compute the reweighted batch gradient $\mathbf{g}_t$ on $B_t$ using equation 9
12:     $\boldsymbol{\theta}_{t+1} \leftarrow \text{OPTIMIZER}(\boldsymbol{\theta}_t, \mathbf{g}_t)$
13: **end for**
    **Return** Finetuned model parameters $\boldsymbol{\theta}_T$

---

**Practical implementation** In practice, we apply the following two techniques to further improve the efficiency of Algorithm 1.

When computing the in-context approximation in equation 6, including the entire holdout set as demonstrations is often infeasible due to prompt length constraints. A straightforward approach is to partition the holdout set into multiple subsets, compute scores for each subset, and average the results; however, this is computationally expensive. Instead, we select the top-$k$ holdout examples most similar to each candidate using $k$-nearest neighbor (kNN) search in an embedding space. In addition, rather than computing scores at every training iteration, we update them only $R$ times periodically over the course of training. Experimental results show that, despite these practical approximations, our method consistently improves alignment performance.

The complete procedure incorporating these techniques is presented as Algorithm 2 in Appendix B.4. We ablate the effects of different choices of $k$ and $R$ in Section 5.3 and analyze the computational overhead of our implementation in Section 5.5.

## 5 EXPERIMENTAL RESULTS

We apply our method to both SFT and preference-based alignment (DPO and SimPO), comparing ICA score-based reweighting to standard training (i.e., without reweighting) and to reweighting using scores computed by baseline methods, across multiple models and datasets.

### 5.1 EXPERIMENTAL SETUP

**Datasets** We consider two data selection scenarios. **High-quality selection** prioritizes expert-level data using a smaller curated holdout set: for SFT, we use Alpaca as training data and sample high-quality holdout examples from its curated version, Alpaca-cleaned (Taori et al., 2023); for preference optimization (DPO and SimPO), we use UltraFeedback-binarized (Cui et al., 2023), which provides preference pairs with scalar quality scores, enabling the construction of high-quality holdout and test sets. **Domain-relevant selection** prioritizes examples relevant to a target domain using a domain-specific holdout set: for SFT, we use Yahoo_Answers_Topics[3], and for preference optimization, we use SHP-2 (Ethayarajh et al., 2022); both datasets contain domain labels, allowing selection of a target domain for evaluating out-of-domain alignment. Dataset split details are provided in Appendix B.1.

---

[3]`https://huggingface.co/datasets/community-datasets/yahoo_answers_topics/viewer?views%5B%5D=train`

Within the high-quality data selection scenario, we additionally evaluate our method on a reasoning-intensive benchmark, GSM8K (Cobbe et al., 2021), which consists of high-quality grade-school math word problems. To test the method's ability to identify high-quality examples in noisy data, a random fraction of chain-of-thought (CoT) examples in the training split are intentionally perturbed. Experimental setup details are provided in Section 5.4, with full results deferred to Appendix C.4.

**Evaluation protocol and metrics** The objective of data selection in this paper is to choose a subset of training data such that a model trained on it minimizes the holdout loss. Accordingly, we evaluate our method by measuring how closely the model's outputs align with the test set targets. To this end, we define the win rate as the percentage of pairwise comparisons in which a response from a model trained with our method is judged more semantically aligned with the target output than that of a model trained with standard (non-reweighted) or baseline methods. Comparisons are performed by GPT-4o (2024-08-06).

In the main experiments, each model is trained and evaluated once due to computational constraints. An analysis of training and evaluation variability is provided in Appendix B.2.

**Models and training** We evaluate our approach on multiple model families and scales, including LLaMA-3-8B-Instruct, LLaMA-3-3B-Instruct, Qwen-3-8B, and Qwen-3-4B. Models are fine-tuned using two parameter-updating paradigms: full-parameter fine-tuning and parameter-efficient LoRA (Hu et al., 2021). Detailed training configurations are provided in Appendix B.3, and LoRA fine-tuning results with our method are reported in Appendix C.1.

**Default setting** By default, we perform full-parameter training and use the ICA score to reweight training examples. We compute ICA scores using the two practical techniques described in Section 4.3 (see Appendix B.4 for additional implementation details). Specifically, for each candidate, we use the top $k = 3$ holdout examples as in-context demonstrations and update scores for all training examples $R = 1$ time (i.e., computing scores for all training examples only at the initialization step $t = 0$). When selecting the top $k$ holdout examples in embedding space, we adopt all-mpnet-base-v2 (Reimers & Gurevych, 2019) to compute embeddings.

**Baselines** In the main experiments, we compare our method with standard training (without reweighting) and two closely related baselines using alternative scores: (1) **RHO-Loss** (Mindermann et al., 2022), which approximates the holdout loss score using a model trained on the holdout set; (2) **One-shot learning** (Li et al., 2023b), which scores each candidate as the difference between the one-shot loss with the candidate included as context and the zero-shot loss without it. Detailed formulas for computing scores with these two methods are provided in Appendix B.5.

In the SFT setting, we further compare against **LESS** Xia et al. (2024) and **GREATS** Wang et al. (2024), two data selection methods based on influence functions. For a fair comparison, we adopt the original selection scheme of each method: when comparing to LESS, we select the top 5% of examples based on precomputed scores before training; when comparing to GREATS, we select the top 50% of examples within each batch for gradient updates. Comparison results are summarized in Section 5.2 and provided in full in Appendix C.3.

## 5.2 MAIN RESULTS

We report the performance of our method across SFT, DPO, and SimPO, comparing against standard training and baseline approaches, in Tables 1, 2, and 3.

**Comparison to standard training** Across all datasets and model families, incorporating our reweighting method leads to consistently better alignment than standard training without reweighting. The improvements hold for both SFT and preference-based alignment, demonstrating that our method provides a robust advantage across different learning paradigms.

**Comparison to baselines** Compared to baseline reweighting methods, our approach consistently outperforms one-shot learning across all settings, with win rates above 50%, and often exceeding 60%. Against RHO-Loss, which approximates the same selection criterion in equation 4 but requires training an auxiliary reference model, our method achieves comparable—and in some cases

Table 1: Win rates (%) of our method against each baseline when applied to SFT across models and datasets (higher values indicate better performance of our method).

| Model | Alpaca | | | Yahoo_Answers_Topics | | |
|---|---|---|---|---|---|---|
| | w/o | RHO-Loss | One-Shot | w/o | RHO-Loss | One-Shot |
| LLaMA 3B | 77.81 | 48.96 | 57.03 | 78.90 | 46.73 | 62.33 |
| LLaMA 8B | 80.55 | 49.92 | 62.11 | 85.10 | 54.03 | 66.93 |
| Qwen 4B | 71.21 | 50.56 | 56.82 | 80.30 | 49.93 | 58.73 |
| Qwen 8B | 82.92 | 51.21 | 58.33 | 82.93 | 54.43 | 63.13 |

Table 2: Win rates (%) of our method against each baseline when applied to DPO across models and datasets (higher values indicate better performance of our method).

| Model | UltraFeedback-binarized | | | StanfordNLP/SHP-2 | | |
|---|---|---|---|---|---|---|
| | w/o | RHO-Loss | One-Shot | w/o | RHO-Loss | One-Shot |
| LLaMA 3B | 61.05 | 46.85 | 57.60 | 79.90 | 56.70 | 60.20 |
| LLaMA 8B | 64.00 | 48.25 | 58.40 | 77.20 | 48.70 | 55.10 |
| Qwen 4B | 64.30 | 51.90 | 54.60 | 79.20 | 49.90 | 57.00 |
| Qwen 8B | 64.85 | 49.90 | 60.45 | 70.40 | 47.60 | 54.40 |

superior—performance, while avoiding the additional cost of reference model training. These results hold for both SFT and preference-based training paradigms and across LLaMA and Qwen models at different scales (3B and 8B).

For SFT on the Yahoo_Answers_Topics dataset, and using the same data selection scheme, our method achieves comparable performance to LESS and GREATS, with win rates slightly above 50% on both 8B LLaMA and Qwen models, while being more computationally efficient (see Appendix C.3 for full results). This efficiency advantage arises because LESS and GREATS rely on local-linearization–based influence functions, whose first-order assumptions can be violated under batch data selection and require additional computation to control approximation error, whereas our method avoids this overhead by deriving scores from a Bayesian formulation rather than a first-order expansion.

## 5.3 ABLATION STUDIES

We conduct ablation studies under the SFT setting on LLaMA-3B-Instruct trained on the Yahoo_Answers_Topics dataset to examine the effect of key design choices in our method on model performance. Results are detailed in Appendix B.6, with key findings presented below.

**Large $k$ is not required for good performance**  To reduce computational cost, we use the top-$k$ holdout examples selected via kNN in embedding space as in-context demonstrations, rather than the full holdout set. Using $k = 3$ as the default, smaller or larger values yield no improvement, with win rates below 50% relative to the default: 43.5% for $k = 1$, 48.0% for $k = 5$, and 46.0% for $k = 10$. The performance drop for $k > 3$ may be due to additional holdout examples being less relevant. This indicates that a small number of holdout examples is sufficient to maintain strong alignment while preserving computational efficiency.

**More frequent score updates improve alignment**  We investigate the effect of the total number of score computations $R$ on model performance. In the default setting, scores are computed once at initialization ($R = 1$). Increasing $R$ to 3, 5, and 9 yields win rates of 50.73%, 52.77%, and 51.6%, respectively. These results suggest that increasing $R$ can further improve alignment, while very frequent updates (e.g., $R = 9$) appear unnecessary, as the dynamics of ICA weights (see Section 5.5) show that the relative importance of training examples stabilize in later stages of training.

Table 3: Win rates (%) of our method against each baseline when applied to simPO across models and datasets (higher values indicate better performance of our method).

| Model | UltraFeedback-binarized | | | StanfordNLP/SHP-2 | | |
|---|---|---|---|---|---|---|
| | w/o | RHO-Loss | One-Shot | w/o | RHO-Loss | One-Shot |
| LLaMA 3B | 62.20 | 47.20 | 55.45 | 93.90 | 50.70 | 55.90 |
| LLaMA 8B | 64.70 | 49.90 | 57.70 | 63.30 | 46.50 | 51.30 |
| Qwen 4B | 64.95 | 47.65 | 51.20 | 66.00 | 46.80 | 49.60 |
| Qwen 8B | 65.55 | 50.60 | 54.35 | 77.10 | 45.50 | 53.60 |

**Filtering is less effective**   As an alternative to our default reweighting approach, we consider percentile-based filtering using ICA scores, retaining only examples above a specific percentile. Simple filtering is generally less effective, with win rates below 50% relative to reweighting. A threshold of the 75th percentile yields a higher win rate (48.67%) than the 50th (40.80%) and 90th (40.07%), indicating that retaining some lower-scoring examples can be beneficial, while too many degrade performance, so the threshold must be chosen carefully. In contrast, adaptive reweighting eliminates this need for manually selecting a filtering threshold by automatically adjusting example importance.

**More advanced embeddings improve performance**   To evaluate the effect of different embedding models for selecting the top-$k$ holdout examples, we test BAAI/bge-m3 (Chen et al., 2024), a stronger embedding model. BAAI/bge-m3 yields higher win rates (52%) compared to the default all-mpnet-base-v2, suggesting that more capable embeddings can further improve the alignment achieved by our method.

## 5.4   ADDITIONAL RESULTS

We present additional results to complement the main evaluations, using the SFT setting as a representative example. Specifically, we report absolute performance metrics that do not rely on a model-based judge and further evaluate the effectiveness of our method on a reasoning-intensive benchmark. We highlight the main findings below, with detailed results deferred to Appendix C.2 and C.4.

**Consistency across metrics**   To complement GPT-judged win rates, we report absolute performance as measured by perplexity (PPL) and BERTScore Zhang et al. (2019) for SFT models trained on Yahoo_Answers_Topics using our reweighting method, compared against each baseline. As shown in Appendix C.2, the performance trends under these absolute metrics are consistent with those observed under GPT-judged win rates.

**Mitigating the effect of noise on GSM8k**   To evaluate our method's ability to select valuable training examples from datasets that may contain suboptimal instances, we conduct SFT on both the original training split and a variant with intentionally corrupted CoT annotations. Details on how the data were corrupted and how the holdout set was constructed are provided in Appendix B.1. Pass@1 accuracy under zero-shot CoT prompting is reported in Table 16 in Appendix C.4, with key findings summarized below.

SFT trained on the original high-quality split achieves the best performance. Using our reweighting method on the corrupted split yields the second-best results, with only a small performance gap. These results show that reweighting noisy training data with our method can achieve performance comparable to training on the original high-quality dataset, highlighting its potential to mitigate the impact of substantial noise, even on complex reasoning tasks.

## 5.5   ANALYSIS

**Computational complexity**   Our method introduces two sources of overhead beyond standard fine-tuning: (i) a one-time precomputation to obtain embeddings, and (ii) periodic score updates.

The latter is dominated by the in-context approximation term, which involves additional forward passes and accounts for most of the computational cost. We measure runtime on four NVIDIA A6000 GPUs (48 GB each) for LLaMA-3B-Instruct fine-tuned on Yahoo_Answers_Topics. Precomputation is negligible, on the order of seconds, and the additional time for score computation and reweighting is reported as a percentage relative to the runtime of standard fine-tuning. Our method adds only ~1.5% overhead, compared to roughly 10% for RHO-Loss and 4% for one-shot learning. Full results are provided in Appendix B.7.

**Score distribution across training examples**  We analyze SFT on Yahoo_Answers_Topic to examine the scores assigned to training examples. Using a holdout set sampled from the Sports topic, we compute average score for examples from each domain (Figure 1). Examples from Sports receive higher scores under both our method and RHO-Loss compared to One-Shot Learning and other topics, demonstrating that our method achieves strong alignment while requiring less computational overhead.

**Dynamics of ICA weights**  We observe that ICA weights adjust dynamically during the training process, with most changes occurring early and stabilizing as training progresses. Using SFT results on Yahoo_Answers_Topics with $R = 5$ as an example, correlations between weights from the first update and subsequent updates are 0.89, 0.75, 0.69, and 0.71. Since the weights stabilize later, very frequent score updates are unnecessary. Additional analyses in Appendix D.2 examine response patterns produced by models trained with our method, and Appendix D.3 presents instruction-response pairs with high and low ICA scores, offering further insight into our method's effectiveness.

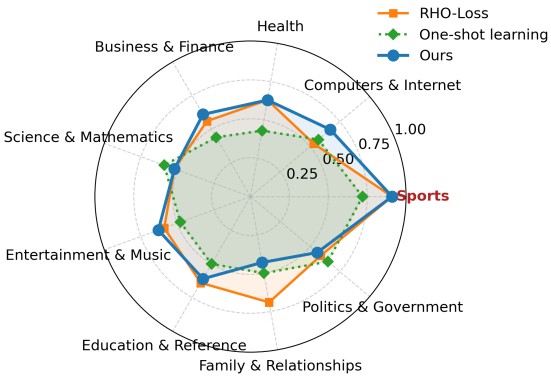

Figure 1: Average scores assigned to training examples from each domain (normalized to $[0, 1]$). The target domain is highlighted in red, and different colors indicate different scoring methods, with our method in blue. Higher scores indicate stronger alignment with the target domain.

## 6  CONCLUSION AND DISCUSSION

We propose a principled, resource-efficient framework for data selection in LLM fine-tuning. Our approach leverages an in-context approximation (ICA) to estimate the holdout loss of the model after including each candidate example in training, without requiring additional finetuning or a reference model. The resulting ICA scores are used to dynamically reweight gradient updates during fine-tuning. Empirical results show that ICA score-based reweighting consistently improves model alignment across SFT, DPO, and SimPO over diverse datasets and backbone architectures, with only marginal computational overhead ($\sim 1.5\%$).

We also note limitations and directions for future work. We focus on settings where a high-quality holdout set is available to estimate the importance of training examples. If the holdout set is noisy or unrepresentative, generalization to unseen data may be impaired, even if alignment appears strong. Additionally, our experiments use off-policy training paradigms. Directly applying ICA to on-policy methods, such as PPO, would require frequent recomputation of scores for all newly generated data, creating a substantial computational bottleneck. Addressing these challenges is a promising direction for future work.

## ETHICS STATEMENT

All authors have read, adhere to, and explicitly acknowledge the ICLR Code of Ethics. This work does not involve human subjects or private user data, and all datasets used in our experiments are publicly available and appropriately cited, with full details of data processing and holdout/test splits provided in the appendix to ensure transparency and reproducibility.

All experimental procedures and analyses adhere to accepted research integrity standards, and we disclose any potential conflicts of interest: none exist. This work does not involve applications that directly pose safety or legal risks beyond standard LLM use.

The assistive use of LLMs in preparing this paper, to refine grammar and improve the clarity of the text, is documented in Appendix E, and the authors take full responsibility for all content.

## REPRODUCIBILITY STATEMENT

All code and materials used in our experiments are available at `https://github.com/microsoft/HeurAgenix/tree/dpo`. To facilitate reproducibility, we provide detailed descriptions of all datasets used, including holdout and test set construction and dataset splits, in Appendix B.1. The algorithm used in our experiments is detailed in Appendix B.4, with full training parameters, LoRA configurations, and evaluation settings provided in Appendix B.3. Prompts used for querying LLMs when computing ICA scores, as well as prompts used for GPT to judge win rates, are provided in Appendix D.1.

For theoretical results, our method builds on the holdout loss expression introduced in Mindermann et al. (2022), and we reproduce the complete derivation in Appendix A. Together, these materials allow others to reproduce our experiments and verify our results.

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

## A  DERIVATION OF EQUATION 3 UNDER A BAYESIAN FRAMEWORK

We reproduce here the derivation from Mindermann et al. (2022) for completeness. Let $\boldsymbol{\theta}_t := \boldsymbol{\theta}^\star(\mathcal{D}_t)$ denote the model trained on $\mathcal{D}_t$, and define the loss as the negative log-likelihood $\ell(\mathbf{y} \mid \mathbf{x}; \boldsymbol{\theta}) = -\log p(\mathbf{y} \mid \mathbf{x}; \boldsymbol{\theta})$. We use $\mathbf{x}^{\text{ho}}$ and $\mathbf{y}^{\text{ho}}$ to denote the collections of inputs and outputs of the holdout examples, respectively.

Under a Bayesian framework, the holdout loss of a model trained with a particular example can be expressed as

$$\log p(\mathbf{y}^{\text{ho}} \mid \mathbf{x}^{\text{ho}}; \mathcal{D}_t \cup (\mathbf{x}, \mathbf{y})) = \log \frac{p(\mathbf{y} \mid \mathbf{x}; \mathbf{x}^{\text{ho}}, \mathbf{y}^{\text{ho}}, \mathcal{D}_t) \, p(\mathbf{y}^{\text{ho}} \mid \mathbf{x}^{\text{ho}}, \mathbf{x}; \mathcal{D}_t)}{p(\mathbf{y} \mid \mathbf{x}, \mathbf{x}^{\text{ho}}; \mathcal{D}_t)}$$

$$= \log \frac{p(\mathbf{y} \mid \mathbf{x}; \mathbf{y}^{\text{ho}}, \mathbf{x}^{\text{ho}}, \mathcal{D}_t) \, p(\mathbf{y}^{\text{ho}} \mid \mathbf{x}^{\text{ho}}; \mathcal{D}_t)}{p(\mathbf{y} \mid \mathbf{x}; \mathcal{D}_t)}$$

$$\propto \ell(\mathbf{y} \mid \mathbf{x}; \boldsymbol{\theta}_t) - \ell(\mathbf{y} \mid \mathbf{x}; \boldsymbol{\theta}^\star(\mathcal{D}_t \cup \mathcal{D}_{\text{ho}}))$$

where the first equality applies Bayes' rule, the second uses a conditional independence assumption, i.e., $p(\mathbf{y}_i \mid \mathbf{x}_i, \mathbf{x}_j, \mathcal{D}_t) = p(\mathbf{y}_i \mid \mathbf{x}_i, \mathcal{D}_t)$, for any $i \neq j$, and the last line drops the candidate-independent term.

## B  DETAILS OF EXPERIMENTS

### B.1  DATASET SPLITS

We provide details on the construction of the training, holdout, and test sets for all datasets used in our experiments.

- **Alpaca and Alpaca-cleaned** (Taori et al., 2023) Alpaca-cleaned is a curated version of Alpaca. The holdout set consists of the first 10,000 examples from Alpaca-cleaned, and the test set consists of the last 10,000 examples. For training, we use all Alpaca examples except those whose corresponding examples in Alpaca-cleaned have been reserved for the test set, resulting in 84,022 training examples.

Table 4: Win rates of our method against each baseline for LLaMA-3-8B-Instruct on Yahoo_Answers_Topics, averaged over three GPT-based evaluations from a single training run, with corresponding standard deviations reported.

| Ours against | w/o | RHO-Loss | One-shot |
|---|---|---|---|
| **Win Rate (%)** | $85.10 \pm 0.30$ | $54.03 \pm 0.20$ | $66.93 \pm 0.50$ |

Table 5: Win rates (%) of our method against each baseline on Yahoo_Answers_Topics, averaged over three independent runs with different random seeds, reported for both LLaMA-3-8B-Instruct and Qwen-3-8B. Standard deviations are also provided for reference.

| Model | w/o | RHO-Loss | One-shot |
|---|---|---|---|
| LLaMA 8B | $82.86 \pm 1.96$ | $49.92 \pm 0.39$ | $62.59 \pm 0.57$ |
| Qwen 8B | $81.67 \pm 1.09$ | $54.23 \pm 0.18$ | $63.81 \pm 0.60$ |

- **Yahoo_Answers_Topics** The holdout and test sets each contain 3,000 examples from the Sports domain. The training set consists of 10,000 examples from each of the remaining domains, together with the holdout examples, totaling 100,300 training examples.

- **UltraFeedback-binarized** (Cui et al., 2023) The dataset is split into a training set (66,282 examples) and a test set (2,000 examples). From the training set, a holdout set of 5,147 pairs is selected, consisting of examples where the chosen response has a quality score $\geq 9$ and the rejected response has a quality score $\geq 7$. All examples in the training set are used for training.

- **StanfordNLP/SHP-2** (Ethayarajh et al., 2022) We use the Baking domain as the target domain. From the official test split, we rank Baking examples by $(\text{score\_A} + \text{score\_B})$, selecting the top 1,000 as the holdout set and the next 1,000 as the test set. For training, we use the official train split, selecting 1,000 examples from each domain and including the holdout set.

- **GSM8K** (Cobbe et al., 2021) This dataset contains 8.5K high-quality grade school math word problems requiring multi-step reasoning. From the training split, we set aside 1,000 examples as a high-quality holdout. The remaining training examples are used as the model training set, of which a randomly selected 40% are intentionally corrupted (e.g., via CoT dropout, shuffled CoT, or CoT re-generation using a weaker model). The original test split is used for evaluation.

In main experiments, the holdout set is included in the training data to ensure fair comparison with baselines that do not use reweighting. We additionally conduct experiments where the holdout set is used only for scoring in Appendix C.5.

## B.2 TRAINING AND EVALUATION VARIABILITY

Each model in the main experiments is trained and evaluated only once. To verify evaluation stability, we repeated the GPT-based evaluation three times for LLaMA-3-8B-Instruct on Yahoo_Answers_Topics. Table 4 reports the average win rates of our method against each baseline, along with standard deviations across these runs. Higher win rates indicate better alignment of our method, and the small standard deviations show that the evaluation results are stable across runs.

To assess training variability, we performed three independent SFT runs with different random seeds for each of the LaMA-3-8B-Instruct and Qwen-3-8B models on Yahoo_Answers_Topics. Table 5 reports the average win rates of our method against each baseline, with standard deviations provided for reference. The mean performance across these training runs aligns closely with the single-run results reported in the main experiments.

Table 6: Detailed training parameters for full fine-tuning, LoRA, and evaluation.

| Full Fine-tuning Parameters | SFT | DPO | SimPO |
|---|---|---|---|
| torch_dtype | bfloat16 | bfloat16 | bfloat16 |
| attn_implementation | flash_attention_2 | flash_attention_2 | flash_attention_2 |
| lr_scheduler_type | cosine | cosine | cosine |
| gradient_accumulation_steps | 16 | 16 | 16 |
| learning_rate | $1 \times 10^{-5}$ | $1 \times 10^{-6}$ | $1 \times 10^{-6}$ |
| max_length | 1024 | - | - |
| max_prompt_length | - | 1024 | 1024 |
| max_seq_length | - | 1024 | 1024 |
| num_train_epoch | 1 | 1 | 1 |
| optim | adamw_torch | adamw_torch | adamw_torch |
| per_device_train_batch_size | 1 | 1 | 1 |
| per_device_eval_batch_size | 4 | 4 | 4 |
| seed | 42 | 42 | 42 |
| warmup_ratio | 0.1 | 0.1 | 0.1 |
| loss_type | nll | sigmoid | sigmoid |
| beta | - | - | 2.5 |
| gamma_beta_ratio | - | - | 0.55 |
| sft_weight | - | - | 0.0 |
| disable_dropout | - | - | True |
| **LoRA Fine-tuning Parameters** | | | |
| learning_rate | $1 \times 10^{-4}$ | $1 \times 10^{-4}$ | $1 \times 10^{-4}$ |
| lora_r | 8 | 8 | 8 |
| lora_alpha | 16 | 16 | 16 |
| lora_dropout | 0.1 | 0.1 | 0.1 |
| lora_target_modules | [q_proj, k_proj, v_proj, up_proj, down_proj, o_proj, gate_proj] | | |
| lora_task_type | CAUSAL_LM | CAUSAL_LM | CAUSAL_LM |
| **Evaluation Parameters** | | | |
| tester | azure_GPT | | |
| api_version | 2025-01-01-preview | | |
| model | gpt-4o_2024-08-06 | | |
| temperature | 0 | | |
| top_p | 0.95 | | |
| seed | None | | |
| max_tokens | 1600 | | |

## B.3 DETAILS OF TRAINING AND EVALUATION CONFIGURATIONS

We summarize the training and testing configurations for full fine-tuning, LoRA fine-tuning, and evaluation in Table 6.

## B.4 IMPLEMENTATION DETAILS OF ALGORITHM 1

We provide details of the two techniques introduced in the main text for improving the computational efficiency of Algorithm 1.

**Selecting in-context demonstrations via kNN**   Instead of using the full holdout set $\mathcal{D}_{\mathrm{ho}}$ for in-context learning, we condition on a smaller, more relevant subset of $\mathcal{D}_{\mathrm{ho}}$ selected using embedding similarity.

Specifically, we first precompute and store embeddings of the inputs for all training and holdout examples:

$$\mathbf{h}_i = f(\mathbf{x}_i, \mathbf{y}_i) \quad \text{for } i = 1, \dots, |\mathcal{D}|,$$

$$\mathbf{h}_i^{\mathrm{ho}} = f(\mathbf{x}_i^{\mathrm{ho}}, \mathbf{y}_i^{\mathrm{ho}}) \quad \text{for } i = 1, \dots, |\mathcal{D}^{\mathrm{ho}}|,$$

where $f$ denotes the embedding function (by default, we use all-mpnet-base-v2 (Reimers & Gurevych, 2019)).

---

**Algorithm 2** Enhanced Algorithm 1 with Computational Efficiency Techniques

---

1: **Input:** Training set $\mathcal{D}$; Holdout set $\mathcal{D}_{\text{ho}}$; Pre-trained model parameters $\boldsymbol{\theta}$; Number of training steps $T$; Total number of score computations $R$; kNN hyperparameter $k$; Embedding function $f$; Batch size $n_B$; Optimizer OPTIMIZER

2: Initialize $\boldsymbol{\theta}_0 \leftarrow \boldsymbol{\theta}$
3: $\mathcal{D}, \mathcal{D}_{\text{ho}} \leftarrow$ PREPROCESSING$(\mathcal{D}, \mathcal{D}_{\text{ho}}, f, k)$
4: **for** $t = 0, \cdots, T - 1$ **do**

5:     **if** $t \bmod \frac{|\mathcal{D}|}{n_B R} = 0$ **then**               // Recompute scores every $F$ steps
6:         **for** $(\mathbf{x}_i, \mathbf{y}_i, \mathbf{h}_i, \texttt{Score}_i) \in \mathcal{D}$ **do**
7:             $C_i^k \leftarrow$ GETDEMONSTRATIONSET$(\mathbf{h}_i, \mathcal{D}_{\text{ho}}, k)$
8:             $\texttt{ConditionalLoss}_i \leftarrow \ell(\mathbf{y}_i \mid \mathbf{x}_i, C_i^k; \boldsymbol{\theta}_t)$
9:             $\texttt{Loss}_i \leftarrow \ell(\mathbf{y}_i \mid \mathbf{x}_i; \boldsymbol{\theta}_t)$
10:            $\texttt{Score}_i \leftarrow \texttt{Loss}_i - \texttt{ConditionalLoss}_i$
11:         **end for**
12:     **end if**

13:     Sample batch $B_t \subset \mathcal{D}$ of size $n_B$
14:     Compute per-example weights within the batch using equation 8
15:     Compute the reweighted batch gradient $\mathbf{g}_t$ on $B_t$ using equation 9
16:     $\boldsymbol{\theta}_{t+1} \leftarrow$ OPTIMIZER$(\boldsymbol{\theta}_t, \mathbf{g}_t)$
17: **end for**
    **Return** Finetuned model parameters $\boldsymbol{\theta}_T$

18: **function** PREPROCESSING$(\mathcal{D}, \mathcal{D}_{\text{ho}}, f, k)$
19:     **for** $(\mathbf{x}_i, \mathbf{y}_i)$ in $\mathcal{D}$ **do**
20:         Compute $\mathbf{h}_i \leftarrow f(\mathbf{x}_i, \mathbf{y}_i)$              // Embedding for training example $i$
21:         Initialize $\texttt{Score}_i \leftarrow 0$
22:     **end for**
23:     Update training set $\mathcal{D} \leftarrow \{(\mathbf{x}_i, \mathbf{y}_i, \mathbf{h}_i, \texttt{Score}_i)\}_{i=1}^{|\mathcal{D}|}$

24:     **for** $(\mathbf{x}_i^{\text{ho}}, \mathbf{y}_i^{\text{ho}})$ in $\mathcal{D}_{\text{ho}}$ **do**
25:         Compute $\mathbf{h}_i^{\text{ho}} \leftarrow f(\mathbf{x}_i^{\text{ho}}, \mathbf{y}_i^{\text{ho}})$         // Embedding for holdout example $i$
26:     **end for**
27:     Update holdout set $\mathcal{D}_{\text{ho}} \leftarrow \{(\mathbf{x}_i^{\text{ho}}, \mathbf{y}_i^{\text{ho}}, \mathbf{h}_i^{\text{ho}})\}_{i=1}^{|\mathcal{D}_{\text{ho}}|}$
28: **end function**

29: **function** GETDEMONSTRATIONSET$(\mathbf{h}, \mathcal{D}_{\text{ho}}, k)$      // Select top $k$ holdout examples using embedding similarity
30:     $C^k \leftarrow \{(\mathbf{x}^{\text{ho}}, \mathbf{y}^{\text{ho}}) \in \mathcal{D}_{\text{ho}} \mid \mathbf{h}^{\text{ho}} \text{ is among the } k \text{ nearest to } \mathbf{h}\}$
31:     **Return** $C^k$
32: **end function**

---

For each candidate $(\mathbf{x}, \mathbf{y}) \in \mathcal{D}$ with embedding $\mathbf{h}$, we select the top-$k$ holdout examples whose embeddings are closest to $\mathbf{h}$ to form a demonstration subset $C^k$. We replace the full $\mathcal{D}_{\text{ho}}$ with this subset when computing the ICA score (line 6 of Algorithm 1). Then the ICA score can be computed as

$$s(\mathbf{x}, \mathbf{y}; \boldsymbol{\theta}_t) \approx \ell(\mathbf{y} \mid \mathbf{x}; \boldsymbol{\theta}_t) - \ell(\mathbf{y} \mid \mathbf{x}, C^k; \boldsymbol{\theta}_t),$$

where

$$C^k := \{(\mathbf{x}^{\text{ho}}, \mathbf{y}^{\text{ho}}) \in \mathcal{D}_{\text{ho}} \mid \mathbf{h}^{\text{ho}} \text{ is among the } k \text{ nearest to } \mathbf{h}\}.$$

Although in our experiments we recompute the kNN search at each scoring step, the demonstration subsets can be precomputed once and reused across iterations to further amortize the cost.

**Periodic score updates** In Algorithm 1, scores are computed at every training step. To improve efficiency, we instead perform score computation only $R$ times during training. At each recomputa-

Table 7: Effect of Top-$k$ on win rates; higher values indicate better performance for the corresponding $k$.

| Top-$k$ | 1 | 5 | 10 |
|---|---|---|---|
| **Win rate (%)** | 43.50 | 48.03 | 46.07 |

Table 8: Effect of total number of score computations $R$ on win rates; higher values indicate better performance for the corresponding $R$.

| **R** | 3 | 5 | 9 |
|---|---|---|---|
| **Win rate (%)** | 50.73 | 52.77 | 51.60 |

tion point, scores for all training examples are updated and stored; in the intervening steps, the most recent scores are reused to determine weights.

The reweighting algorithm incorporating these two techniques is presented in Algorithm 2, where the score update frequency is determined from the training set size, batch size, and the total number of score computations $R$.

### B.5 IMPLEMENTATION DETAILS OF BASELINES

We provide additional details on how scores are computed for each reweighting baseline. These scores are used in place of the ICA score within our reweighting framework. For the subset selection methods LESS (Xia et al., 2024) and GREATS (Wang et al., 2024), we reimplement their publicly available code and follow their original data selection procedures: selecting the top 5% before training for LESS, and 50% of each batch for gradient updates for GREATS.

- **RHO-Loss** (Mindermann et al., 2022) approximates the holdout loss score in equation 4 by replacing the second term with a separate model trained once on the holdout set. For each candidate, the resulting score is

$$s_{\text{RHO-Loss}}(\mathbf{x}, \mathbf{y}) = \ell(\mathbf{y} \mid \mathbf{x}; \boldsymbol{\theta}_t) - \ell(\mathbf{y} \mid \mathbf{x}; \boldsymbol{\theta}^\star(\mathcal{D}_{\text{ho}})).$$

  To replicate this method, we train the target model on the holdout set $\mathcal{D}_{\text{ho}}$ to obtain $\boldsymbol{\theta}^\star(\mathcal{D}_{\text{ho}})$.

- **One-shot learning** (Li et al., 2023b) computes a score for each candidate as the difference between the one-shot loss with the candidate included as context and the zero-shot loss without it:

$$s_{\text{one-shot}}(\mathbf{x}, \mathbf{y}) = \mathcal{L}(\mathcal{D}_{\text{ho}}; \boldsymbol{\theta}_0) - \mathcal{L}(\mathcal{D}_{\text{ho}} \mid (\mathbf{x}, \mathbf{y}); \boldsymbol{\theta}_0)$$

  where, for consistency with our setting, we use the pretrained model $\boldsymbol{\theta}_0$ to perform this one-shot evaluation and compute losses on the holdout set $\mathcal{D}_{\text{ho}}$ instead of the predefined subtasks used in the original paper.

### B.6 RESULTS OF ABLATION STUDIES

Tables 7–10 summarize the ablation studies on LLaMA-3B-Instruct trained on Yahoo_Answers_Topics. Win rates indicate the percentage of responses preferred over the default setting (i.e., $k = 3$, $R = 1$, using all-mpnet-base-v2 as the embedding model). Each table corresponds to varying one design dimension: Top-$k$ (Table 7); total number of score computations $R$ during training (Table 8); percentile threshold for filtering (Table 9); and embedding model (Table 10). Higher win rates indicate better performance of the corresponding design choice; 50% corresponds to parity with the default.

Table 9: Effect of percentile threshold for filtering on win rates; higher values indicate better performance for the corresponding percentile. Here, $\geq x$ indicates retaining examples with scores above the $x$-th percentile.

| Filtering $\geq x$ | 50 | 75 | 90 |
|---|---|---|---|
| Win rate (%) | 40.80 | 48.67 | 40.07 |

Table 10: Win rates under a stronger embedding model against the default; higher values indicate better performance for the stronger model.

| Embedding Model | Win rate (%) |
|---|---|
| BAAI/bge-m3 | 52.13 |

## B.7 COMPUTATIONAL OVERHEAD

We report the computational overhead of our method compared to baseline methods. Table 11 presents the runtime for embedding precomputation, score computation, and total additional runtime relative to standard fine-tuning (computed as additional time divided by the runtime of standard fine-tuning). We train LLaMA-3B-Instruct on Yahoo_Answers_Topics using the default settings of our method, and measure runtime on four NVIDIA A6000 GPUs (48 GB each).

## C ADDITIONAL EXPERIMENTS

### C.1 ADDITIONAL RESULTS USING LORA

To assess the effectiveness of our method across different parameter updating paradigms, we conduct LoRA training using LLaMA-3-8B-Instruct. Table 12 reports win rates for LoRA with our reweighting method relative to regular LoRA training across various alignment methods and datasets. Each value represents the percentage of responses judged closer to the target by GPT-4o_2024-08-06, with higher values indicating better performance. The results demonstrate that our method consistently improves alignment, even under the LoRA parameter updating setting.

### C.2 ABSOLUTE METRICS ON YAHOO_ANSWERS_TOPICS

Table 13 reports perplexity (PPL) and BERTScore for SFT models trained on Yahoo_Answers_Topics under different reweighting methods. Lower PPL and higher BERTScore indicate better model performance.

### C.3 COMPARISON WITH LESS AND GREATS

Tables 14 and 15 report win rates of our method against LESS and GREATS under the same data selection paradigm for SFT on `Yahoo_Answers_Topics`, together with PPL and BERTScore. Comparison results on GSM8k are reported in the following section, using Pass@1 accuracy. Runtime of computing scores for all training examples using each data selection method is provided in Table 17.

### C.4 RESULTS ON GSM8K

Table 16 reports Pass@1 accuracy for SFT models trained on the original and corrupted splits of GSM8k using our reweighting method and baseline approaches.

Table 11: Runtime for embedding precomputation and score computation (seconds), and total additional runtime relative to standard training (percentage). Higher values indicate greater computational cost.

| Metric | Ours | RHO-Loss | One-shot |
|---|---|---|---|
| **Precomputation** (s) | 2 | 5400 | 2 |
| **Score Computation** (s) | 800 | 800 | 2000 |
| **Additional Runtime** (%) | 1.5 | 10 | 4 |

Table 12: Win rates of LoRA trained with our method relative to LoRA trained without our method using LLaMA-3-8B-Instruct. Higher values indicate better performance.

| Alignment Method | SFT | | DPO | | SimPO | |
|---|---|---|---|---|---|---|
| | Alpaca | Yahoo | UltraFeedback | SHP-2 | UltraFeedback | SHP-2 |
| **Win rate (% ↑)** | 71.55 | 68.13 | 61.23 | 57.33 | 56.03 | 60.13 |

Table 13: Absolute-metric performance of SFT-trained models on Yahoo_Answers_Topics under different reweighting strategies (including no reweighting), evaluated using perplexity (PPL) and BERTScore. Lower PPL and higher BERTScore indicate better model performance. Best and second-best results are highlighted in bold and underlined, respectively.

| | LLaMA 8B | | Qwen 8B | |
|---|---|---|---|---|
| **Method** | PPL (↓) | BERTScore (↑) | PPL (↓) | BERTScore (↑) |
| **Ours** | **11.23** | **0.84** | 10.30 | 0.87 |
| w/o | 16.71 | 0.80 | 13.04 | 0.82 |
| RHO-Loss | 11.49 | **0.84** | **10.00** | **0.88** |
| One-shot | 14.92 | 0.82 | 12.02 | 0.82 |

## C.5 EXCLUDING HOLDOUT EXAMPLES FROM TRAINING

In our main experiments, the holdout set is included in training to ensure a fair comparison with baselines that do not use reweighting, which otherwise have no access to holdout examples. To examine the potential impact of including holdout examples on performance, we reran SFT on the Yahoo_Answers_Topic dataset, keeping the holdout set strictly for scoring. Results are presented in Table 18. Even without training on the holdout set, our method outperforms the standard baseline and one-shot learning, although it performs slightly below RHO-Loss, which trains a reference model on the holdout set. Nevertheless, our method remains comparable to RHO-Loss while using holdout examples only as in-context demonstrations for scoring.

## C.6 PERFORMANCE WHEN USING THE HOLDOUT SET ALONE

To show that test performance relies on the broader training data and the holdout set alone does not suffice, we train the model solely on the holdout set (see results in Table 19). As expected, its performance is lower than our method trained on the full training set with ICA reweighting, highlighting the benefit of combining broader training data with data selection guided by the holdout set.

## D ADDITIONAL ANALYSIS

### D.1 PROMPTS FOR IN-CONTEXT APPROXIMATION AND MODEL EVALUATION

Table 20 shows the prompts we use. The first section provides a standard query, the second includes the query with holdout examples as in-context demonstrations used for computing ICA scores via

Table 14: Comparison with additional baselines under the SFT setting on Yahoo_Answers_Topics using LLaMA-3-8B. Win rates are computed within the same selection paradigm. Lower PPL and higher BERTScore indicate better performance. Best values are shown in bold.

| Method | Win rate (%) (Ours vs Baseline) | PPL ($\downarrow$) | BERTScore ($\uparrow$) |
|---|---|---|---|
| *Online selection* | | | |
| GREATS (50% per batch) | 52.03 | 22.07 | 0.78 |
| Ours (50% per batch) | – | **22.01** | **0.79** |
| *Pre-training selection* | | | |
| LESS (Top 5%) | 51.40 | 20.55 | **0.80** |
| Ours (Top 5%) | – | **18.91** | **0.80** |

Table 15: Comparison with additional baselines under the SFT setting on Yahoo_Answers_Topics using Qwen-3-8B. Win rates are computed within the same selection paradigm. Lower PPL and higher BERTScore indicate better performance. Best values are shown in bold.

| Method | Win rate (%) (Ours vs Baseline) | PPL ($\downarrow$) | BERTScore ($\uparrow$) |
|---|---|---|---|
| *Online selection* | | | |
| GREATS (50% per batch) | 50.91 | **19.70** | **0.80** |
| Ours (50% per batch) | – | 19.74 | **0.80** |
| *Pre-training selection* | | | |
| LESS (Top 5%) | 50.90 | 17.24 | 0.77 |
| Ours (Top 5%) | – | **17.03** | **0.78** |

Table 16: Pass@1 accuracy (%) on GSM8k under zero-shot CoT prompting for LLaMA-3-8B and Qwen-3-8B, trained with SFT using different data selection strategies. Best and second-best results are highlighted in bold and underlined, respectively.

| Method | LLaMA-3-8B | Qwen-3-8B |
|---|---|---|
| *Without reweighting* | | |
| Pretrained | 75.28 | 80.52 |
| SFT (original) | **81.91** | **83.27** |
| SFT (corrupted) | 52.75 | 62.40 |
| *Reweighting-based methods* | | |
| RHO-Loss | 79.50 | 81.16 |
| One-shot | 72.42 | 76.19 |
| **Ours (reweighting)** | 79.95 | 83.04 |
| *Pre-training selection* | | |
| LESS (Top 5%) | 77.63 | 80.81 |
| **Ours (Top 5%)** | 78.92 | 80.81 |
| *Online selection* | | |
| GREATS (50% per batch) | 80.03 | 80.50 |
| **Ours (50% per batch)** | 80.18 | 80.49 |

ICA approximation, and the third shows prompts used for evaluation, where GPT judges which candidate response is closer to the target output.

## D.2 RESPONSE LENGTH AS AN ALIGNMENT INDICATOR

Our method effectively captures the characteristics of the holdout dataset. In Yahoo_Answers_Topics, which consists of daily conversational data collected from internet users, the

Table 17: Score computation time (in seconds) on LLaMA-3-8B for our method, LESS, and GREATS when training on Yahoo_Answers_Topics with SFT. Times for LESS and GREATS are based on our reimplementation. For LESS, the two numbers correspond to influence score computation and periodic model retraining to address the circular dependence between data selection and model updates.

| Dataset | Ours | LESS | GREATS |
|---|---|---|---|
| Yahoo_Answers_Topics | 808 | 1,394 + 1,800 | 1,004 |
| GSM8K | 117 | 310 + 900 | 704 |

Table 18: Win rates of our method against each baseline reweighting method for models trained with SFT on `Yahoo_Answers_Topics`, using $\mathcal{D}_{\mathrm{ho}}$ as a strict holdout for scoring (not included in training). Higher win rates indicate better performance of our method. PPL and BERTScore are also reported, with the best and second-best values indicated in bold and underlined, respectively.

| | LLaMA-3-8B | | | Qwen-3-8B | | |
|---|---|---|---|---|---|---|
| Method | Win rate (%) (Ours vs Baseline) | PPL ($\downarrow$) | BERTScore ($\uparrow$) | Win rate (%) (Ours vs Baseline) | PPL ($\downarrow$) | BERTScore ($\uparrow$) |
| w/o | 89.27 | 26.77 | 0.72 | 87.49 | 22.44 | 0.80 |
| RHO-Loss | 47.29 | **12.08** | **0.81** | 49.21 | **12.94** | 0.85 |
| One-shot | 71.33 | 19.45 | 0.79 | 67.02 | 18.73 | 0.82 |
| Ours | – | 14.05 | **0.81** | – | 13.29 | **0.86** |

Table 19: Comparison of our method against the baseline model trained only on the holdout set $\mathcal{D}_{\mathrm{ho}}$ for SFT on Yahoo_Answers_Topics. Win rates indicate the percentage of examples where our method is preferred over the baseline; higher values indicate better performance of our method. PPL and BERTScore are also reported. with best values indicated in bold.

| | LLaMA-3-8B | | | Qwen-3-8B | | |
|---|---|---|---|---|---|---|
| Method | Win rate (Ours vs Baseline) | PPL ($\downarrow$) | BERTScore ($\uparrow$) | Win rate (%) (Ours vs Baseline) | PPL ($\downarrow$) | BERTScore ($\uparrow$) |
| $\mathcal{D}_{\mathrm{ho}}$-only | 58.13 | 12.40 | 0.82 | 56.80 | 11.92 | 0.82 |
| Ours | – | **11.23** | **0.84** | – | **10.30** | **0.87** |

Sports category contains notably shorter responses, with an average token length of 62.07 compared to 86.01 in other categories. Moreover, the proportion of extremely short answers (token length $\leq 5$) is substantially higher in the Sports category (16%) than in others (4%). The model trained using our approach reflects these patterns, producing similarly concise responses to sports-related questions, as shown in Table 21.

### D.3 EXAMPLES OF HIGH- AND LOW-SCORING RESPONSES

To provide intuition for the effectiveness of our scoring method, we select two example pairs and present them in Table 22. In each pair, the prompt is the same, but the responses differ: one is from Alpaca and the other from Alpaca-cleaned. For each response, we show the score assigned by our method. We observe that responses receiving higher scores tend to exhibit clearer structure, more complete coverage of the instruction's intention, and more precise or domain-relevant content, while low-scoring responses often display redundancy, incomplete answers, or missing key details. These examples illustrate that our method prioritizes clearer structure and precise content, helping explain why it improves model alignment.

Table 20: Prompts for ICA computation and model evaluation

**Query**

```
You are an expert assistant.
Answer the following question:
{question}
```

**Query with In-Context Demonstrations**

```
You are an expert assistant.  Follow the examples:
Q: {example_question_1}
A: {example_answer_1}
Q: {example_question_2}
A: {example_answer_2}
...
Answer the following question:
{question}
```

**Evaluation**

```
You are an expert assistant.  Given a question, a standard answer,
and two candidate answers, indicate which candidate is closer to the
standard.
Instructions:
    1. Choose the candidate closer to the standard, not necessarily the
       higher quality.
    2. Consider content, length, and style relative to the standard
       answer.
    3. Be concise; output only a JSON object with the winner.
Question:  {question}
Standard answer:  {standard_answer}
Candidate 1:  {candidate_answer_1}
Candidate 2:  {candidate_answer_2}

Output format:
If candidate 1 is closer:  {"winner":  1}
If candidate 2 is closer:  {"winner":  2}
```

Table 21: Example responses from the Sports category showing alignment of model outputs with target domain response length.

**Response Example**

```
Question:  Are there any sports where it's an advantage to be little?
Answer from dataset:  A Jockey.
Answer from model after SFT: Yes, I would say that wrestling is one of
them.
Answer from model after Ours+SFT: Polo.
Answer from LLaMA-3B-Instruct before fine-tuning:  While being tall
can be an advantage in some sports, there are indeed some sports
where being shorter can be beneficial.  Examples include:  gymnastics,
wrestling, boxing, table tennis, fencing, weightlifting, and cycling.
Shorter athletes can benefit from a lower center of gravity, faster
movement, or easier maneuverability, though technique, strength, and
endurance remain crucial.
```

# E  USE OF LARGE LANGUAGE MODELS

In adherence to ICLR 2026 policy, we disclose the assistive use of Large Language Models (LLMs) in the preparation of this paper. The models were employed to refine grammar and improve the

clarity of the text. The authors have reviewed all LLM-generated contributions and take full responsibility for the content and integrity of this work.

Table 22: Example pairs from Alpaca and Alpaca-cleaned with scores assigned by our method

**Example 1**

Instruction:  Analyze the following document and group the topics it
talks about.

Response (score = 1.99):  Uber being a leading company in the
ride-sharing industry since its launch in 2009.  It has disrupted
the transportation industry, creating a business model in which it
acts as a middleman between two parties:  the rider and the driver.
Uber's main service offering includes a mobile app and website, which
allows riders to request a ride from an available driver.  This ride
can be paid for using Uber's payment system, which was developed in
partnership with a variety of third-party processors.  The company also
offers various add-on services, such as \UberPool", which allows riders
to share a ride with other riders to save money, and \UberRush", which
allows riders to schedule a ride for a future time.

Response (score = 7.45):  Topic 1:  Ride-sharing industry
Topic 2:  Uber's business model
Topic 3:  Mobile App/Website
Topic 4:  Payment System
Topic 5:  Add-on services (UberPool & UberRush)

**Example 2**

Instruction:  Describe the types of facial recognition technology used
in security systems.

Response (score = 5.56):  Facial recognition technology is used in
various security systems to identify and verify individuals based
on their facial features.  Some common types of facial recognition
technologies used in security systems include:
1.  2D Facial Recognition:  This is the most commonly used facial
recognition technology, where the system captures a 2D image of a
person's face and compares it with the images in its database to find a
match.
2.  3D Facial Recognition:  This technology uses 3D sensors to capture
the shape and contours of a person's face, providing a more accurate
and reliable identification than 2D recognition.
3.  Thermal Facial Recognition:  This technology uses thermal imaging
to capture the heat patterns that are unique to each individual's face.
This technology is particularly useful in low-light conditions or where
the subject is wearing items that obscure their face.
4.  Skin Texture Analysis:  This advanced facial recognition technology
captures the unique patterns and lines on a person's skin to identify
and verify their identity.
These are some of the common types of facial recognition technologies
that are used in security systems, and they offer varying levels of
accuracy and reliability in identifying individuals.

Response (score = 15.40):  Facial recognition technology used in
security systems can come in many forms, such as biometrics-based
authentication, 3-D face recognition, or the use of algorithms to
detect facial features.  Biometrics-based authentication requires users
to provide physical attributes such as fingerprints, iris scans, and
facial recognition for access.  3-D face recognition is more secure and
uses more than two-dimensional image recognition to identify a person.
Algorithms can detect the shape, size, and position of facial features,
allowing for more accurate verification.

