# OpenReview forum: "Holdout-Loss-Based Data Selection for LLM Finetuning via In-Context Learning"
_ICLR.cc/2026/Conference — ICLR 2026 Poster_

### Official Review · Reviewer_HKFV · 2025-10-27

**Soundness:** 2
**Presentation:** 2
**Contribution:** 3
**Rating:** 6
**Confidence:** 2

**Summary:**

This paper presents a framework to select and re-weight training examples during LLM fine-tuning based on In Context Approximation (ICA) of holdout loss. The ICA score measures (by proxy) a training example's contribution to minimizing the holdout loss for one step. The authors improve the algorithm by sampling the top k most similar holdout examples (to a training example) and updating scores at intervals. Their results show improvement in win rate (alignment performance) across SFT, DPO and SimPO fine-tuning settings.

**Strengths:**

1. This algorithm uses in-context learning as a data re-weighting tool to improve alignment-based finetuning - which I find very interesting and seems practically efficient (only ~1.5% overhead reported).
2. Experiments with SFT, DPO and SimPO suggest generalizability across different paradigms. Their LLM-as-a-judge metric shows a clear empirical signal to support the utility of the in-context approximation. Particularly liked the experiment in Figure 1 -  the analysis furthers support for their work and has far-reaching implications in domain adaptation.
3. Smart optimizations backed by ablation experiments further the case for in-context approximation to select high-valued samples for LLM alignment.

**Weaknesses:**

1. This paper needs to be compared with other off-policy re-weighting baselines like GREATS [1].
2. Reweighting stability across different runs is unclear. Another ablation experiment that could support the paper would be if this method aids in faster convergence.


[1] GREATS: Online Selection of High-Quality Data for LLM Training in Every Iteration

**Questions:**

1. While the paper focuses on selecting samples for alignment, it would be interesting if the method affects overall LLM performance.
2. Did you consider comparison with other domain adaptation / alignment baselines?
3. This paper will be strengthened with a re-weighting sensitivity analysis to determine over multiple runs (at least a couple runs in case of compute restrictions) that the same set of examples are highly valued (generally measured by the standard deviation of data value across runs).

---

> ### Author Response · Authors · 2025-11-25
> **Responses to Reviewer HKFV Part 1**
>
> We thank the reviewer for the insightful questions and suggestions. We provide detailed responses below. In all experiments involving our method, we use the default settings unless stated otherwise. For brevity, we refer to LLaMA-3-8B-Instruct as Llama3-8B, and Qwen-3-8B as Qwen3-8B throughout the responses.
>
> **Weakness 1: Compare with other reweighting baseline**
>
> We will include comparisons to GREATS in the revised version. At this stage, we report results with LESS because it has substantially lower computational overhead. Like GREATS, LESS relies on influence-function–based scoring, but with improved efficiency. Specifically, it selects the top 5% of training examples based on the computed scores, whereas our method uses scores to reweight all gradient updates. To make the comparison fair, LESS scores were applied under both reweighting and top-5% selection, with results shown in Tables 4 and 5, respectively. ICA scores were applied using the same schemes.
>
> When LESS scores are used for reweighting, the method achieves performance comparable to ours but is substantially more computationally intensive: computing scores for all training examples on the Yahoo dataset takes about 3 hours, whereas our method requires only 20 minutes.
>
> Using LESS's top-5% selection improves efficiency, but performance in terms of PPL and BERTScore can decrease, as expected due to the reduced training data. Applying ICA scores for top-5% data selection achieves slightly better results than LESS.
>
> This suggests the potential of our method: it can achieve improved alignment by leveraging all training data through reweighting, while maintaining computational efficiency.
>
>
> [1] Less: Selecting influential data for targeted instruction tuning.
>
>
>
> **Table 4. SFT results on `Yahoo_Answers_Topic` with our reweighting method compared to LESS. Both methods use computed scores for reweighting. PPL and BERTScore for our method are averaged over 3 runs.**
>
>
> **Llama3-8B**
>
> | Method             | Win rates of ours against  (%, ↑) | PPL ↓                 | BERTScore (F) ↑        |
> |--------------------|-------------------------------------|------------------------|--------------------------|
> | **Ours, reweighting**           | –                                   | 11.23   | 0.84       |
> | LESS, reweighting               | 52.01                              | 11.86                 | 0.84                     |
>
>
>
> **Qwen3-8B**
>
> | Method             | Win rates of ours against  (%, ↑) | PPL ↓                 | BERTScore (F) ↑        |
> |--------------------|-------------------------------------|------------------------|--------------------------|
> | **Ours, reweighting**           | –                             | 10.30   | 0.87       |
> | LESS, reweighting               | 49.20                          | 11.23                 | 0.87    |
>
>
> **Table 5. SFT results on `Yahoo_Answers_Topic` with our reweighting method compared to LESS. Both methods use computed scores to select the top 5% of training data.**
>
>
> **Llama3-8B**
>
> | Method             | Win rates of ours against  (%, ↑) | PPL ↓                 | BERTScore (F) ↑        |
> |--------------------|-------------------------------------|------------------------|--------------------------|
> | **Ours, top 5%**           | –                                   | 18.91   | 0.80     |
> | LESS, top 5%              | 51.40                            | 20.55                | 0.80                |
>
>
>
> **Qwen3-8B**
>
> | Method             | Win rates of ours against  (%, ↑) | PPL ↓                 | BERTScore (F) ↑        |
> |--------------------|-------------------------------------|------------------------|--------------------------|
> | **Ours, top 5%**           | –                             | 17.03   | 0.78       |
> | LESS, top 5%               | 50.90                        | 17.24                 | 0.77   |
>
>
> **Weakness 1 & Question 3: Reweighting stability across different runs**
>
>
> To examine the stability of reweighting, we looked at it from two angles:
>
> 1. Across runs with different training seeds (after the first score update).
> In our setup, the first score update happens right after initialization. Since the initialized model is the same in our experiments, the $k$ holdout examples selected for each training example are also the same. As a result, the ICA scores after the first update are identical across runs.
>
> 2. Within a single run, across different score-update rounds.
> We also looked at how weights evolve over the course of training. Using SFT results on the `Yahoo_Answers_Topic` dataset as an example, for the same set of data points, the correlations between weights from the first update and subsequent updates are 0.89, 0.75, 0.69, and 0.71, respectively.
>
> This shows that most changes happen early in training, and the weights gradually stabilize, suggesting that ICA can adjust weights dynamically while remaining relatively stable throughout training.

---

> ### Author Response · Authors · 2025-11-25
> **Responses to Reviewer HKFV Part 2**
>
> **Weakness 1: Ablation on convergence speed.**
>
> We thank the reviewer for the suggestion. To examine whether our method aids faster convergence, we will include a figure in the revision showing the training loss of SFT (cross-entropy loss) over iterations. For `GSM8K` , we observe that reweighting can achieve lower training loss within the same number of iterations. In particular, our method achieves a convergence speed similar to RHO-Loss and LESS, and is much faster than one-shot learning, while attaining a lower final training loss than all baselines. In addition, it appears to be more stable than LESS or SFT without reweighting.
>
> For the`Yahoo_Answers_Topic` dataset, since our goal is to better align with the target domain (sports) rather than improving answer accuracy as in `GSM8K` , reweighting does not significantly affect convergence speed or the final loss value.
>
>
>
>
> **Question 1:  While the paper focuses on selecting samples for alignment, it would be interesting if the method affects overall LLM performance.**
>
>
>  We agree that it is important to understand whether our method also benefits overall LLM performance. In particular, we focus on the model's reasoning capabilities, and conduct experiments experiments on the `GSM8K` [2] dataset, which contains 8.5K high-quality grade school math word problems requiring multi-step reasoning. We used 1,000 examples from the training split as a high-quality holdout and the remainder as training data, of which 40% were intentionally corrupted (e.g., via CoT dropout, shuffled CoT, or CoT re-generation using a weaker model). Models were trained with SFT using each reweighting method and evaluated on the test split. Accuracies are reported for zero-shot CoT @1 in Table 1.
>
> The results show that reweighting the corrupted training data with our method (*italic*) can achieve performance comparable to training directly on the original high-quality dataset (**bold**). This suggests that our reweighting approach not only shows promise in mitigating the impact of substantial noise in the training data, but may also help the model maintain its reasoning capabilities despite noisy inputs.
>
> [2] Training Verifiers to Solve Math Word Problems
>
> **Table 1: Zero-shot CoT @1 accuracies on `GSM8K` for Llama3-8B and Qwen3-8B (%)**
>
> | Model                      | Llama3-8B | Qwen3-8B  |
> | -------------------------- | --------- | --------- |
> | Pretrained                 | 75.28     | 80.52     |
> | SFT (original train split) | **81.91** | **83.27** |
> | SFT (corrupted data)       | 52.75     | 62.40     |
> | Ours  (corrupted data)     | *79.95*   | *83.04*   |
> | RHO  (corrupted data)      | 79.50     | 81.16     |
> | One-shot  (corrupted data) | 72.42     | 76.19     |
>
>
> We also examined the weights assigned to data points of different quality by our method. We find that our method tends to assign higher weights to the original high-quality data and lower weights to corrupted examples:
>
> | Data Quality | Ratio (% of total data) | Weight assigned |
> |------------|-------|---------------|
> | Original | 40%  | 0.82          |
> | CoT dropped | 10%  | 0.08          |
> | CoT shuffled | 10%  | 0.46          |
> | Numbers replaced | 10% | 0.21 |
> | CoT re-generated | 30% | 0.48 |
>
> This further supports the potential of our method to mitigate the impact of noise in the training data.
>
>
> **Question 2: Consider comparison with other domain adaptation / alignment baselines.**
>
> We thank the reviewer for this question.  In our experiments, we compare training with our method to training without reweighting for DPO and SimPO,  which are two commonly used alignment methods. The results suggest that our method can help improve the performance of these standard alignment approaches.
>
> We would be happy to consider additional baselines if the reviewer has specific suggestions.

---

> > ### Author Response · Authors · 2025-11-28
> >
> > I hope you’re well. I wanted to follow up to see if you had a chance to review our rebuttal. We posted updates addressing your main points, including comparisons to additional reweighting/selection baselines, stability and convergence analyses, and broader evaluations beyond alignment, and we responded to your specific questions. If any further details, figures, or ablations would help your assessment, we’d be happy to provide them promptly. Thank you again for your thoughtful review and time.

---

### Official Review · Reviewer_t4k1 · 2025-10-29

**Soundness:** 2
**Presentation:** 3
**Contribution:** 2
**Rating:** 4
**Confidence:** 4

**Summary:**

This paper proposes a new framework for data selection during the fine-tuning of LLMs. The core problem addressed is the negative impact of noisy or off-target data often present in large fine-tuning datasets. The proposed solution aims to identify and prioritize high-value training examples based on their potential to improve the model's performance on a small, high-quality holdout set. The key innovation is the In-Context Approximation (ICA) method, which leverages the LLM's in-context learning (ICL) ability to efficiently estimate the holdout loss change associated with training on a candidate example, without requiring expensive retraining or auxiliary models. These ICA scores are then used to dynamically reweight gradient updates during fine-tuning across various paradigms like SFT, DPO, and SimPO. The authors present experimental results suggesting that this method consistently improves model alignment compared to standard training.

**Strengths:**

1. The core idea of using ICL to approximate the impact of a training sample on holdout loss (the ICA score) is innovative and computationally efficient. It cleverly avoids the high costs associated with traditional data valuation methods that require retraining or reference models, potentially making holdout-loss-based selection more practical.


2. The method is motivated by theoretical considerations, building upon the holdout loss approximation framework established in prior work (e.g., RHO-Loss). Linking the ICA score to a first-order update towards the holdout optimum provides a principled justification for its use.




3. The framework demonstrates versatility by showing positive results across multiple fine-tuning techniques (SFT, DPO, SimPO) , different model families and sizes (LLaMA-3, Qwen-3; 3B/8B) , and varied data selection scenarios (quality improvement, domain relevance).

**Weaknesses:**

1. The theoretical justification for the core In-Context Approximation (ICA) itself appears weak. While the goal (approximating holdout loss impact) is theoretically grounded, the paper does not provide a rigorous analysis or proof showing why or how accurately a single forward pass with k-shot ICL (Eq. 6) approximates the complex model state change ($\theta^{*}(\mathcal{D}_{t}\cup\mathcal{D}_{ho})$ in Eq. 4) that involves actual gradient updates. The reliance on the general observation that ICL performs "implicit fine-tuning" feels insufficient to bridge this gap, weakening the claim of a fully "theoretically grounded" method.

2. The empirical evaluation relies heavily on a potentially unreliable and limited metric, hindering convincing validation. The primary evaluation metric is the "win rate" determined by GPT-4o judging pairwise comparisons. This methodology is susceptible to the biases and limitations of the judge model, costly, hard to reproduce, and lacks granularity. The absence of standard, objective task-specific metrics (e.g., accuracy on downstream tasks, perplexity, benchmark scores like MT-Bench/AlpacaEval) makes it difficult to assess the actual impact on model capabilities beyond alignment style as perceived by one specific LLM.

3. The experimental scope feels somewhat narrow, potentially limiting the generalizability of the findings. While multiple models and methods are tested, the experiments primarily use only two main dataset pairs (Alpaca/Alpaca-cleaned for quality, Yahoo/SHP for domain). Validation on a wider range of dataset types, sizes, and noise levels would be necessary to confirm the robustness and broad applicability claimed. Furthermore, the comparison is limited to RHO-Loss and One-Shot learning; comparing against simpler heuristics (e.g., loss-based filtering, perplexity) or more recent data selection methods could provide better context.

4. This paper would be strengthened by a comparison of this method to other recent frameworks for data selection and robust optimization during fine-tuning [1, 2, 3].

[1] Provably Robust DPO: Aligning Language Models with Noisy Feedback. ICML 2024.

[2] ROPO: Robust Preference Optimization for Large Language Models. ICML 2025.

[3] LESS: Selecting influential data for targeted instruction tuning. ICML 2024.

**Questions:**

1. The ablation study shows that using k=3 for ICL examples performs best, with performance degrading for k=5 and k=10. This seems counter-intuitive, as ICL performance often benefits from more examples. Could the authors provide some intuition or analysis for this observation?

2. How sensitive is the method to the choice of the k holdout examples selected via kNN? Does the variance introduced by different kNN selections significantly affect the stability of the ICA scores and the final performance?

3. There seems to be an inconsistency in model naming: the text mentions using LLaMA-3-3B-Instruct and Qwen-3-4B , but Table 3 refers to "Qwen 3B". Please clarify if a 3B Qwen model was used or if this is a typo for 4B.

---

> ### Author Response · Authors · 2025-11-25
> **Responses to Reviewer t4k1 Part 1**
>
> We thank the reviewer for the insightful questions and suggestions. We provide detailed responses below. In all experiments involving our method, we use the default settings unless stated otherwise. For brevity, we refer to LLaMA-3-8B-Instruct as Llama3-8B, and Qwen-3-8B as Qwen3-8B throughout the responses.
>
>
> **Weakness 1: Theoretical justification for ICA is weak.**
>
> We agree that the current justification of ICA relies on the intuition that in-context learning can be interpreted as performing an implicit parameter update. A fully formal derivation quantifying the accuracy of this approximation is nontrivial and left for future work; therefore, ICA is theoretically grounded, though not fully so.
>
> To avoid any potential confusion, we have removed the phrase "theoretically grounded" and revised the abstract as follows:
>
> > We present a principled, resource-efficient framework for data selection and reweighting.
>
>
> **Weakness 2:  Absence of standard, objective task-specific metrics**
>
> To provide complementary evaluation metrics, we reran SFT on the  `Yahoo_Answers_Topic`dataset with LLaMA 8B and Qwen 8B three times using different random seeds with our reweighting method. Along with GPT-4o–judged win rates, we report perplexity (PPL) and BERTScore [1] as complementary metrics.
>
> As shown in Table 1, our method continues to outperform the no-reweighting and one-shot baselines under these additional metrics and remains comparable to the more computation-intensive RHO-Loss baseline.
>
>
>
> [1] Bertscore: Evaluating text generation with bert.
>
>
> **Table 1. SFT results on `Yahoo_Answers_Topic` over 3 runs with different seeds. Win rates of baselines against our method, as well as PPL and BERTScore of our method, are averaged over 3 runs.**
>
>
> **Llama3-8B**
>
> | Method             | Win rates of ours against (%, ↑) | PPL ↓                 | BERTScore (F) ↑        |
> |--------------------|-------------------------------------|------------------------|--------------------------|
> | SFT w/o reweighting        | 82.86              | 16.71                 | 0.80                     |
> | **Ours**           | –                                   | 11.23   | 0.84          |
> | RHO                | 49.92              | 11.49                 | 0.84                     |
> | One-shot           | 62.59              | 14.92                 | 0.82                     |
>
>
> **Qwen3-8B**
>
> | Method             | Win rates of ours against (%, ↑) | PPL ↓                 | BERTScore (F) ↑        |
> |--------------------|-------------------------------------|------------------------|--------------------------|
> | SFT w/o reweighting        | 81.67           | 13.04                 | 0.82                     |
> | **Ours**           | –                             | 10.30   | 0.87         |
> | RHO                | 54.23           | 10.00                 | 0.88                     |
> | One-shot           | 63.81           | 12.02                 | 0.82                     |
>
> **Weakness 3 The experimental scope feels somewhat narrow.**
>
> We agree that broader evaluation is important to demonstrate robustness. To this end, we conducted experiments on the `GSM8K` [2] dataset, which contains 8.5K high-quality grade school math word problems requiring multi-step reasoning. We used 1,000 examples from the training split as a high-quality holdout and the remainder as training data, of which 40% were intentionally corrupted (e.g., via CoT dropout, shuffled CoT, or CoT re-generation using a weaker model). Models were trained with SFT using each reweighting method and evaluated on the test split. Accuracies are reported for zero-shot CoT @1 in Table 2.
>
> The results show that reweighting the corrupted training data with our method (*italic*) can achieve performance comparable to training directly on the original high-quality dataset (**bold**). This suggests that our reweighting approach not only shows promise in mitigating the impact of substantial noise in the training data, but may also help the model maintain its reasoning capabilities despite noisy inputs.
>
> [2] Training Verifiers to Solve Math Word Problems
>
> **Table 1: Zero-shot CoT @1 accuracies on `GSM8K` for Llama3-8B and Qwen3-8B (%)**
>
> | Model                      | Llama3-8B | Qwen3-8B  |
> | -------------------------- | --------- | --------- |
> | Pretrained                 | 75.28     | 80.52     |
> | SFT (original train split) | **81.91** | **83.27** |
> | SFT (corrupted data)       | 52.75     | 62.40     |
> | Ours  (corrupted data)     | *79.95*   | *83.04*   |
> | RHO  (corrupted data)      | 79.50     | 81.16     |
> | One-shot  (corrupted data) | 72.42     | 76.19     |

---

> ### Author Response · Authors · 2025-11-25
> **Responses to Reviewer t4k1 Part 2**
>
> We also examined the weights assigned to data points of different quality by our method. We find that our method tends to assign higher weights to the original high-quality data and lower weights to corrupted examples:
>
> | Data Quality | Ratio (% of total data) | Weight assigned by our method |
> |------------|-------|---------------|
> | Original | 40%  | 0.82          |
> | CoT dropped | 10%  | 0.08          |
> | CoT shuffled | 10%  | 0.46          |
> | Numbers replaced | 10% | 0.21 |
> | CoT re-generated | 30% | 0.48 |
>
> This further supports the potential of our method to mitigate the impact of noise in the training data.
>
>
>
> **Weakness 3 Comparison is limited; Weakness 4: Comparison with other recent data selection work.**
>
> For comparison, we also evaluate LESS [3], which leverages a high-quality reference set for reweighting but uses an influence-function-based method. LESS uses its computed scores to select the top 5% of training data, whereas our method uses scores to reweight all gradient updates. To make the comparison fair, LESS scores were applied under both reweighting and top-5% selection, with results on `Yahoo_Answers_Topic` shown in Tables 2 and 3, respectively. ICA scores were applied using the same schemes.
>
> When LESS scores are used for reweighting, the method achieves performance comparable to ours but is substantially more computationally intensive: computing scores for all training examples on the Yahoo dataset takes about 3 hours, whereas our method requires only 20 minutes.
>
> Using LESS's top-5% selection improves efficiency, but performance in terms of PPL and BERTScore can decrease, as expected due to the reduced training data. Applying ICA scores for top-5% data selection achieves slightly better results than LESS.
>
> This suggests the potential of our method: it can achieve improved alignment by leveraging all training data through reweighting, while maintaining computational efficiency.
>
>
> [3] Less: Selecting influential data for targeted instruction tuning.
>
>
> **Table 2. SFT results on `Yahoo_Answers_Topic` with our reweighting method compared to LESS. Both methods use computed scores for reweighting. PPL and BERTScore for our method are averaged over 3 runs.**
>
>
> **Llama3-8B**
>
> | Method             | Win rates of ours against  (%, ↑) | PPL ↓                 | BERTScore (F) ↑        |
> |--------------------|-------------------------------------|------------------------|--------------------------|
> | **Ours, reweighting**           | –                                   | 11.23   | 0.84       |
> | LESS, reweighting               | 52.01                              | 11.86                 | 0.84                     |
>
>
>
> **Qwen3-8B**
>
> | Method             | Win rates of ours against  (%, ↑) | PPL ↓                 | BERTScore (F) ↑        |
> |--------------------|-------------------------------------|------------------------|--------------------------|
> | **Ours, reweighting**           | –                             | 10.30   | 0.87       |
> | LESS, reweighting               | 49.20                          | 11.23                 | 0.87    |
>
>
> **Table 3. SFT results on `Yahoo_Answers_Topic` with our reweighting method compared to LESS. Both methods use computed scores to select the top 5% of training data.**
>
>
> **Llama3-8B**
>
> | Method             | Win rates of ours against  (%, ↑) | PPL ↓                 | BERTScore (F) ↑        |
> |--------------------|-------------------------------------|------------------------|--------------------------|
> | **Ours, top 5%**           | –                                   | 18.91   | 0.80     |
> | LESS, top 5%              | 51.40                            | 20.55                | 0.80                |
>
>
>
> **Qwen3-8B**
>
> | Method             | Win rates of ours against  (%, ↑) | PPL ↓                 | BERTScore (F) ↑        |
> |--------------------|-------------------------------------|------------------------|--------------------------|
> | **Ours, top 5%**           | –                             | 17.03   | 0.78       |
> | LESS, top 5%               | 50.90                        | 17.24                 | 0.77   |
>
> **Question 1:  It seems counter-intuitive that using k=3 for ICL examples performs best.**
>
> We appreciate the observation. Our setting focuses on alignment rather than standard in-context learning: in-context demonstrations are used to score the relative importance of a candidate for reducing holdout loss, so more examples are not necessarily better. In particular, the top holdout examples are the most semantically similar, and additional examples tend to be less relevant, which likely explains the performance drop for $k > 3$.

---

> > ### Author Response · Authors · 2025-11-25
> > **Responses to Reviewer t4k1 Part 3**
> >
> > **Question 2: Sensitivity to the choice of the k holdout examples selected via kNN.**
> >
> > We thank the reviewer for the question. We hope we understand it correctly: if the question refers to whether the same k holdout examples are selected each time for a given training example, then under the same embedding model, the selection is fixed and would not change. However, using a different embedding model may change the set of k holdout examples for the same training example. For instance, as shown in Table 9 of the main manuscript, switching from `all-mpnet-base-v2` to the stronger embedding model `BAAI/bge-m3` further improves alignment performance, likely because the stronger embedding model provides more accurate semantic similarity.
> >
> >
> >
> > **Question 3: Inconsistency in model naming in Table 3**
> >
> > We thank the reviewer for pointing out this inconsistency. This was a typo: the model we use is Qwen-3-4B, and the correct abbreviation is Qwen 4B. We have corrected this in the revised manuscript.

---

> > > ### Author Response · Authors · 2025-11-28
> > >
> > > I hope you’re well. I wanted to follow up to check whether you’ve had a chance to review our rebuttal. We posted updates addressing your main points, including clarifying the theoretical framing, adding complementary objective metrics and broader experiments, expanding baseline comparisons, and responding to questions on k, kNN selection, and the naming inconsistency. If any additional details, figures, or ablations would help your assessment, we’d be happy to provide them promptly. Thank you again for your thoughtful review and time.

---

### Official Review · Reviewer_zAys · 2025-10-29

**Soundness:** 2
**Presentation:** 3
**Contribution:** 2
**Rating:** 4
**Confidence:** 3

**Summary:**

The paper proposes In‑Context Approximation (ICA) to estimate, per training example, how much the holdout loss would decrease if the example were used for training.  It defines an ICA score as the difference between unconditional and conditional losses and uses these scores to reweight per-example gradients. The method targets SFT, DPO, and SimPO and is evaluated on LLaMA-3 (3B/8B) and Qwen-3 (4B/8B) across Alpaca/Alpaca‑cleaned, Yahoo Answers Topics (domain shift), UltraFeedback‑binarized, and SHP‑2. Results are reported as GPT‑4o judged win rates of  vs. standard training (no reweighting),  RHO‑Loss, and One‑Shot Learning.

**Strengths:**

- Simple, general mechanism: Framing per-example value via an in‑context estimate of “training on the holdout set” is elegant and easy to drop into standard training loops (Eq. 6–9; Alg. 1).

- Broad applicability: Demonstrated on SFT, DPO, SimPO and across two model families/sizes. Tables 1–3 show “Ours vs. w/o reweighting” win rates generally >60% and often much higher (e.g., LLaMA‑8B SFT Yahoo: 85.10%), supporting cross‑paradigm usefulness.

- Low overhead claim with ablations: The method adds ~1.5% runtime (Table 10), and ablations on k (few in‑context demos), R (score recomputations), weighting vs. filtering, and embedding choice are included (Sec. 5.3; Tables 6–9), which helps practitioners calibrate cost/performance trade‑offs.

- Clear exposition of pipeline: Pseudocode (Alg. 1; Alg. 2) and prompts (Table 12) make the approach easy to reimplement; training/eval config is summarized (Table 5)

**Weaknesses:**

1. Apparent holdout leakage in multiple setups: In Yahoo Answers Topics, the paper states: “The holdout and test sets each contain 3,000 Sports examples. The training set consists of 1,000 examples from each of the remaining domains, together with the holdout examples, totaling 12,000 training examples.” (Appendix B.1, emphasis added). This means the holdout set is inside the training set, which undermines the notion of “holdout loss,” confounds the ICA scoring (which conditions on $D_{ho}$), and risks overstating gains. A similar pattern occurs for SHP‑2 and UltraFeedback‑binarized (“From the training set, a holdout set … is selected… All examples in the training set are used for training”). These choices likely cause data leakage between “scorer” (holdout) and “trainer,” conflating reweighting with simply learning from the curated subset directly.
This affects both internal validity (does ICA help when the holdout is truly held out?) and comparability to RHO‑Loss/One‑Shot under proper splits.

2. Evaluation design relies on single‑seed training and model‑judge only: The paper explicitly trains and evaluates each model only once due to compute constraints (Sec. 5.1). Variability is assessed only by repeating GPT‑based evaluation (Table 4), not by rerunning training. This leaves uncertainty about training variance and statistical significance of reported win‑rates. Moreover, the main metric is pairwise win rate judged by GPT‑4o (temperature 0, seed None; Table 5/“Evaluation Parameters”), with no human evaluation or confidence intervals.


3. Theoretical grounding for ICA is informal The abstract claims: “Under a local linearization, ICA is equivalent to a first‑order update toward the holdout optimum” (p. 1), but the paper does not present a derivation or bounds for this equivalence. Appendix A reproduces the RHO‑Loss Bayesian derivation but does not analyze the error of the ICA approximation or its dependence on 𝑘, context formatting, or model scale. This weakens the “theoretically grounded” claim.

4. Comparisons and reporting: Absolute performance is not reported; only win rates vs. baselines. Without absolute metrics (e.g., accuracy, reward, BLEU, or task‑specific scores), it is hard to gauge practical effect size. Baselines exclude several strong data‑selection methods (e.g., influence‑function variants, Grad‑Match, importance resampling) beyond citations. Given the paper’s emphasis on efficiency, a compute‑matched subset of these would strengthen claims.

5. Dynamic reweighting is emphasized, yet the default setting computes scores once at t=0 (R=1; Sec. 5.1), which effectively turns the method into a static ranking for most experiments; the ablation (increasing R) shows only modest gains (Table 7).

**Questions:**

1. Holdout leakage: Why are holdout examples included in the training set (Appendix B.1, Yahoo/SHP‑2/UltraFeedback)? Please re‑run the core experiments with strictly disjoint train/holdout/test splits and report results, especially for Yahoo‑Sports and UltraFeedback.

2. Theory gap: Can you provide the promised derivation (or at least a formal sketch with assumptions) showing the local‑linearization equivalence between ICA and a first‑order update toward the holdout optimum? What conditions on model smoothness / context length /
k are required?

3. Variance and significance: Please run ≥3 training seeds for at least one representative setting per paradigm (SFT/DPO/SimPO) and report mean ± std and binomial confidence intervals for win rates. Also include absolute scores on standard benchmarks where available.

4. Dynamic vs. static: Since the default is R=1, how much do results change for R>1 under a fixed compute budget (i.e., reducing training steps to keep total GPU hours constant)?

5. Prompt sensitivity: How sensitive is ICA to demonstration formatting and ordering? Can you include a robustness table varying the in‑context template / example order?

## Suggestion:

- Ablate proper splits: Re‑report Tables 1–3 with strictly disjoint holdout (for scoring) and training datasets, and add a variant where you train on the holdout alone to show that ICA’s gains are not due simply to including curated data.

- Absolute metrics: In addition to judge‑based win rates, include task‑specific metrics or reward m- odel scores to contextualize effect sizes.

- Compute‑fair baseline suite: Add at least one influence‑based (or Grad‑Match / importance‑resampling) baseline under matched compute, even on a reduced scale, to strengthen comparative claims.

- Report overhead vs. R: Extend Table 10 to show overhead for R=3/5/9 to back the claim of efficiency when more frequent updates are used (Table 7).  Also see https://openreview.net/pdf?id=p0KTYl2B9T for overhead scaling.

- Clarify claims: Either include the local‑linearization derivation or soften language from “equivalent” to “motivated by,” and discuss limitations (e.g., prompt length constraints and ICA’s off‑policy nature mentioned in Sec. 6)

---

> ### Author Response · Authors · 2025-11-25
> **Responses to Reviewer zAys Part 1**
>
> We thank the reviewer for the insightful questions and suggestions. We provide detailed responses below. In all experiments involving our method, we use the default settings unless stated otherwise. For brevity, we refer to LLaMA-3-8B-Instruct as Llama3-8B, and Qwen-3-8B as Qwen3-8B throughout the responses.
>
>
>
> **Weakness 1, Question 1, Suggestion 1 on Apparent holdout leakage**
>
> We include the holdout set in training to ensure a fair comparison with baselines that do not use reweighting, which would otherwise have no access to holdout examples, while reweighting methods use them for scoring.
>
> To address concerns about holdout leakage,  we reran SFT on the `Yahoo_Answers_Topic` dataset with LLaMA 8B and Qwen 8B, using $\mathcal{D}$ for training while keeping $\mathcal{D}_{\text{ho}}$ strictly as a holdout set for scoring (See Table 1).  Even without including the holdout set in training, our method still outperforms both the baseline without using reweighting and one-shot learning, and remains comparable to the more computationally intensive RHO-Loss.
>
> **Table 1. SFT results on `Yahoo_Answers_Topic`, using $\mathcal{D}$ for training and $\mathcal{D}_{\text{ho}}$ as a strict holdout. PPL↓ and BERTScore↑ are also reported.**
>
> **Llama3-8B**
>
> | Method        | Win rates of ours against (%, ↑) | PPL ↓ | BERTScore (F) ↑   |
> |---------------|------------------|-------|--------------|
> |  SFT w/o reweighting  | 89.27           | 26.77 | 0.72         |
> | **Ours**      | –                | 14.05 | 0.81         |
> | RHO           | 47.29           | 12.08 | 0.81         |
> | One-shot      | 71.33           | 19.45 | 0.79         |
>
> **Qwen3-8B**
>
> | Method        | Win rates of ours against  (%, ↑) | PPL ↓   | BERTScore (F) ↑   |
> |---------------|-----------|-------|-----------|
> | SFT w/o reweighting   | 87.49    | 22.44 | 0.80      |
> | **Ours**      | –         | 13.29 | 0.86      |
> | RHO           | 49.21    | 12.94 | 0.85      |
> | One-shot      | 67.02    | 18.73 | 0.82      |
>
> **Weakness 2, Weakness 4, Question 3 on single‑seed training and model‑judge metrics only; Absolute performance is not reported; Report mean, std and CI**
>
>  To evaluate training variability, we reran SFT on the `Yahoo_Answers_Topic` dataset with LLaMA 8B and Qwen 8B three times using different random seeds with our reweighting method (see Table 2).  The mean, standard deviation, and binomial confidence interval of win rates of ours against each baseline are reported in Table 3. The results across three runs indicate that our method is robust to random seed.
>
> In addition to reporting win rates against each baseline judged by GPT-4o (2024-08-06), we also report perplexity (PPL) and BERTScore[1] to provide complementary evaluation metrics (see Table 2). The results under these additional metrics are consistent with the win-rate comparisons.
>
>
> [1] Bertscore: Evaluating text generation with bert.
>
>
> **Table 2. SFT results on `Yahoo_Answers_Topic` over 3 runs with different seeds. Win rates of baselines against our method, as well as PPL and BERTScore of our method, are reported for all 3 runs.**
>
>
> **Llama3-8B**
>
> | Method             | Win rates of ours against (%, ↑) | PPL ↓                 | BERTScore (F) ↑        |
> |--------------------|-------------------------------------|------------------------|--------------------------|
> | SFT w/o reweighting        | 85.10, 82.04, 81.44              | 16.71                 | 0.80                     |
> | **Ours**           | –                                   | 11.18, 11.08, 11.42   | 0.84, 0.84, 0.83         |
> | RHO                | 49.93, 50.30, 49.52              | 11.49                 | 0.84                     |
> | One-shot           | 62.11, 62.43, 63.22              | 14.92                 | 0.82                     |
>
> **Qwen3-8B**
>
> | Method             | Win rates of ours against  (%, ↑) | PPL ↓                 | BERTScore (F) ↑        |
> |--------------------|-------------------------------------|------------------------|--------------------------|
> | SFT w/o reweighting        | 82.93, 81.00, 81.09           | 13.04                 | 0.82                     |
> | **Ours**           | –                             | 10.12, 10.55, 10.20   | 0.88, 0.87, 0.87         |
> | RHO                | 54.43, 54.20, 54.07           | 10.00                 | 0.88                     |
> | One-shot           | 63.13, 64.22, 64.09           | 12.02                 | 0.82                     |

---

> > ### Author Response · Authors · 2025-11-25
> > **Responses to Reviewer zAys Part 2**
> >
> > **Table 3. Mean, std, and 95% binomial CI of our win rates against baselines across 3 runs.**
> >
> > **Llama3-8B**
> >
> > | Method             | Mean | Std                 | 95% binomial CI       |
> > |--------------------|-------------------------------------|------------------------|--------------------------|
> > | SFT w/o reweighting        | 82.86             | 1.96        | [0.29, 1.00]                     |
> > | RHO                | 49.92              | 0.39              | [0.01, 0.91]                   |
> > | One-shot           | 62.59              | 0.57                 | [0.29, 1.00]                    |
> >
> > **Qwen3-8B**
> >
> > | Method             | Mean | Std                 | 95% binomial CI       |
> > |--------------------|-------------------------------------|------------------------|--------------------------|
> > | SFT w/o reweighting        | 81.67         | 1.09                 |         [0.29, 1.00]          |
> > | RHO                | 54.23         | 0.18                | 0.88                     | [0.29, 1.00]
> > | One-shot           | 63.81         | 0.60                |  [0.29, 1.00]                  |
> >
> >
> >
> > **Weakness 3， Question 2， Suggestion 5: Theoretical grounding for ICA is informal**
> >
> > We realize that the original abstract claim may have been phrased too strongly and could cause misunderstanding. The intuition, inspired by influence-function-based local linearization, is that maximizing the ICA score roughly corresponds to taking a first-order step toward the holdout optimum, since examples with higher ICA scores contribute most to reducing the holdout loss.
> >
> >
> > However, this is not a formal equivalence. To avoid overstating the claim, we will revise the abstract to:
> >
> > > "Under the Bayesian framework, the example with the highest ICA score approximately corresponds to the one that maximally reduces the holdout loss."
> >
> > We provide a detailed derivation of this revised claim in the updated manuscript.
> >
> >
> >
> > **Weakness 4: Baselines exclude several strong data-selection methods**
> >
> > For comparison, we also evaluate LESS [4], which leverages a high-quality reference set for reweighting but uses an influence-function-based method. LESS uses its computed scores to select the top 5% of training data, whereas our method uses scores to reweight all gradient updates. To make the comparison fair, LESS scores were applied under both reweighting and top-5% selection, with results shown in Tables 4 and 5, respectively. ICA scores were applied using the same schemes.
> >
> > When LESS scores are used for reweighting, the method achieves performance comparable to ours but is substantially more computationally intensive: computing scores for all training examples on the Yahoo dataset takes about 3 hours, whereas our method requires only 20 minutes.
> >
> > Using LESS's top-5% selection improves efficiency, but performance in terms of PPL and BERTScore can decrease, as expected due to the reduced training data. Applying ICA scores for top-5% data selection achieves slightly better results than LESS.
> >
> > This suggests the potential of our method: it can achieve improved alignment by leveraging all training data through reweighting, while maintaining computational efficiency.
> >
> >
> > [4] Less: Selecting influential data for targeted instruction tuning.
> >
> >
> >
> > **Table 4. SFT results on `Yahoo_Answers_Topic` with our reweighting method compared to LESS. Both methods use computed scores for reweighting. PPL and BERTScore for our method are averaged over 3 runs.**
> >
> >
> > **Llama3-8B**
> >
> > | Method             | Win rates of ours against  (%, ↑) | PPL ↓                 | BERTScore (F) ↑        |
> > |--------------------|-------------------------------------|------------------------|--------------------------|
> > | **Ours, reweighting**           | –                                   | 11.23   | 0.84       |
> > | LESS, reweighting               | 52.01                              | 11.86                 | 0.84                     |
> >
> >
> >
> > **Qwen3-8B**
> >
> > | Method             | Win rates of ours against  (%, ↑) | PPL ↓                 | BERTScore (F) ↑        |
> > |--------------------|-------------------------------------|------------------------|--------------------------|
> > | **Ours, reweighting**           | –                             | 10.30   | 0.87       |
> > | LESS, reweighting               | 49.20                          | 11.23                 | 0.87    |

---

> ### Author Response · Authors · 2025-11-25
> **Responses to Reviewer zAys Part 3**
>
> **Table 5. SFT results on `Yahoo_Answers_Topic` with our reweighting method compared to LESS. Both methods use computed scores to select the top 5% of training data.**
>
>
> **Llama3-8B**
>
> | Method             | Win rates of ours against  (%, ↑) | PPL ↓                 | BERTScore (F) ↑        |
> |--------------------|-------------------------------------|------------------------|--------------------------|
> | **Ours, top 5%**           | –                                   | 18.91   | 0.80     |
> | LESS, top 5%              | 51.40                            | 20.55                | 0.80                |
>
>
>
> **Qwen3-8B**
>
> | Method             | Win rates of ours against  (%, ↑) | PPL ↓                 | BERTScore (F) ↑        |
> |--------------------|-------------------------------------|------------------------|--------------------------|
> | **Ours, top 5%**           | –                             | 17.03   | 0.78       |
> | LESS, top 5%               | 50.90                        | 17.24                 | 0.77   |
>
>
>
> **Weakness 5:  Dynamic reweighting is emphasized, but scores are computed once;  increasing R shows only modest gains.**
>
>
> In our experiments, we use $R=1$ as default setting due to computational constraints. However, the weights do evolve over the course of training.
> Using SFT results on the `Yahoo_Answers_Topic` dataset with $R=5$ as an example, for the same set of data points, the correlations between weights from the first update and subsequent updates are 0.89, 0.75, 0.69, and 0.71, respectively.
>
> These results show that most changes occur early, after which the weights gradually stabilize, indicating that ICA can adjust weights dynamically. Although increasing $R=5$ does not lead to substantial gains, this can be seen as a practical advantage: since performance does not heavily depend on frequent updates, only a few score updates are needed, which saves computation.
>
>
>
> **Question 1 on Holdout Leakage**： Please see Table 1.
>
>
> **Question 2 on Theory gap**： Please refer to our response above for a related discussion.
>
>
> **Question 3 on variance and significance**： Please see Table 2 and 3.
>
>
>
> **Question 4 Increase R with compute budget fixed**：
>
>
> We experiment with increased $R$ while keeping the total training time—including both score computation and fine-tuning—roughly constant (about 15 hours) for SFT on the `Yahoo_Answers_Topic` dataset using Llama3-8B-Instruct. To maintain the same compute budget, the number of training batches was reduced as $R$ increased. The results are reported in Table 6.
>
> As $R$ increases to 3, 5, and 7, the win rates relative to $R=1$ continue to improve, even under a fixed compute budget (which means fewer training batches). When $R$ reaches 9, the win rates relative to $R=1$ drop slightly, likely because the large reduction in training batches reduces the benefit of more frequent score updates. On the other hand, this also indicates that frequent score updates are not strictly necessary for maintaining strong performance.
>
>
> **Table 6: SFT results on  `Yahoo_Answers_Topic`  with  Llama3-8B-Instruct for increasing R under a fixed compute budget. Win rates are computed relative to R=1.**
>
>
> | R  | Batches Used / Total Batches | Win rate (%, ↑) | PPL ↓    | BERTScore (F) ↑ |
> |----|---------------------|----------------------|------|-----------|
> | 1  | 1610 / 1610         | -                     | 11.18| 0.84      |
> | 3  | 1510 / 1610         | 50.6                     | 11.69| 0.85      |
> | 5  | 1410 / 1610         | 51                     | 11.63| 0.85      |
> | 7  | 1310 / 1610         | 52.2                     | 11.40| 0.85      |
> | 9  | 1210 / 1610         | 49.4                     | 11.18| 0.85      |
>
>
>
>
> **Question 5 Sensitivity to demonstration formatting and ordering**:
>
>
> We agree that changing demonstration formatting and ordering may affect performance, but we use a standard in-context template [2,3] that was not chosen to favor our method. In the revision, we will include an ablation study on different demonstration formatting and example ordering.
>
>
> [2] Language models are few-shot learners.
> [3] PaLM: Scaling Language Modeling with Pathways

---

> > ### Author Response · Authors · 2025-11-25
> > **Responses to Reviewer zAys Part 4**
> >
> > **Suggestion 1: Train on the holdout alone to ablate the effect of using curated data.**
> >
> > In the main manuscript, all methods use the holdout set in training, so ICA's performance gains are less likely due to including curated examples; to support this, we also include results for the model trained solely on the holdout set of  `Yahoo_Answers_Topic` data in Table 6.
> >
> >
> > **Table 6: SFT results for the model trained solely on the holdout set of  `Yahoo_Answers_Topic` data. Values in parentheses are results using ICA-based reweighting, averaged over 3 runs.**
> >
> > | Model     | Win rate of ours against (%, ↑) | PPL (↓)       | BERTScore (↑) |
> > | --------- | --------------------------------- | ------------- | ------------- |
> > | Llama3-8B | 58.13                             | 12.40 (11.23) | 0.82 (0.84)   |
> > | Qwen3-8B  | 56.80                             | 11.92 (10.30) | 0.82 (0.87)   |
> >
> >
> >
> > **Suggestion 2: Absolute metrics**: Please see Table 2.
> >
> >
> > **Suggestion 3: Compute‑fair baseline suite**: Please see Table 4 and 5.
> >
> >
> > **Suggestion 4: Report overhead vs. R**
> >
> > We report the overhead for a single score update in Table 10 of the manuscript. For multiple updates, the total overhead for score updating scales approximately linearly with $R$. While $R=9$ incurs more computation, frequent updates are not necessary: our ablation shows that $R=1$ or $3$ already provides substantial alignment improvement. The larger $R$ was used only to explore the sensitivity to this hyperparameter.

---

> > > ### Author Response · Authors · 2025-11-28
> > >
> > > I hope you’re well. I wanted to follow up to see if you had a chance to review our rebuttal. We posted updates addressing your main concerns, including proper disjoint splits to avoid holdout leakage, variance and absolute metrics, the theoretical framing and wording, dynamic vs. static scoring under a fixed compute budget, prompt sensitivity, and a compute‑fair baseline comparison with overhead reporting. We also added clarifications where suggested. If any additional details, figures, or reruns would help your assessment, we’d be glad to provide them promptly. Thank you again for your thoughtful review and time.

---

### Official Review · Reviewer_tdGW · 2025-11-01

**Soundness:** 2
**Presentation:** 3
**Contribution:** 2
**Rating:** 4
**Confidence:** 4

**Summary:**

This paper introduces In-Context Approximation (ICA), a resource-efficient framework for data selection and reweighting when fine-tuning Large Language Models. At its core, the method estimates the value of a training example by measuring how it would reduce the loss on a small, curated holdout set. Crucially, this estimation is performed using the model's in-context learning capabilities. By dynamically reweighting gradient updates based on these ICA scores, the method prioritizes high-value data, leading to improved model alignment with minimal computational overhead.

**Strengths:**

+ A major advantage of the ICA method is its ability to approximate the holdout loss (similar to RHO-Loss) without the significant overhead of training a separate reference model on the holdout set. This makes it a more practical and scalable solution.

+ The method demonstrates consistently strong results across various models, datasets, and alignment tasks (SFT, DPO, SimPO). Its performance is shown to be comparable to the more resource-intensive RHO-Loss baseline, validating its effectiveness.

**Weaknesses:**

- The framework's success is dependent on having a high-quality, demonstration holdout set to act as a proxy for the desired test distribution. The performance could degrade if the holdout set is noisy, small, or unrepresentative of the target domain.

- The experiments focus primarily on instruction-following and preference alignment tasks. The paper does not evaluate the method on more complex reasoning domains like mathematics or science, where data quality and logical consistency are critical.

- The method introduces new hyperparameters that may require careful tuning. The paper shows that k=3 (the number of in-context examples) works well for their experiments. However, the optimal k could be highly dependent on the model's context window size, the complexity of the task, and the nature of the domain. This adds an extra layer of tuning complexity for practitioners looking to adopt the method.

- The entire method is built on the assumption that a model's behavior when conditioned on a few examples in-context (ICL) is a faithful approximation of how its parameters would change after a gradient-based update on those same examples (fine-tuning). However, for large SFT training, I am not sure if ICL can mimic the benefit from SFT.

**Questions:**

+ In the description of the demonstration set C (line 114), what does the function s(...) represent? It appears to be a formatting function but is not explicitly defined.
+ Could a direct comparison of the ICA scores and the resulting data rankings be provided against those computed by the RHO-Loss method? This would offer deeper insight into how closely ICA's efficient approximation mirrors the baseline's more expensive calculation.
+ How sensitive is the method's performance to the quality of the holdout examples used for in-context demonstration? It would be insightful to see an ablation study where demonstrations are drawn from random training examples instead of a curated, high-quality set, especially in scenarios where the training and test data share a similar distribution.
+ Could this data selection method be effectively applied to fine-tuning tasks that require complex reasoning, such as solving math problems or answering scientific questions?
+ The derivation for the holdout loss approximation in Equation 3 relies on a conditional independence assumption. This seems to imply that adding a single example (x, y) does not significantly alter the model's posterior. How valid is this assumption in practice, especially since adding more examples to the context is known to change the model's output?

---

> ### Author Response · Authors · 2025-11-25
> **Responses to Reviewer tdGW Part 1**
>
> We thank the reviewer for the insightful questions and suggestions. We provide detailed responses below. In all experiments involving our method, we use the default settings unless stated otherwise. For brevity, we refer to LLaMA-3-8B-Instruct as Llama3-8B, and Qwen-3-8B as Qwen3-8B throughout the responses.
>
>
>
> **Weakness 1**: Performance may degrade if the holdout set is noisy, small, or unrepresentative.
>
> We agree that the effectiveness of our method depends on the holdout set being reasonably representative of the target distribution. In our work, we assume access to a small curated subset, which is often more practical to obtain than a fully clean large dataset. However,  if the holdout set is noisy or unrepresentative, alignment performance may degrade. Developing approaches that remain robust under such conditions is an interesting direction for future work.
>
>
> **Weakness 2 & Question 4**:  The paper does not evaluate the method on more complex reasoning domains like mathematics or science, where data quality and logical consistency are critical.
>
>
>  We agree that evaluation on reasoning-intensive tasks is important. To this end, we conducted experiments on the `GSM8K` [1] dataset, which contains 8.5K high-quality grade school math word problems requiring multi-step reasoning. We used 1,000 examples from the training split as a high-quality holdout and the remainder as training data, of which 40% were intentionally corrupted (e.g., via CoT dropout, shuffled CoT, or CoT re-generation using a weaker model). Models were trained with SFT using each reweighting method and evaluated on the test split. Accuracies are reported for zero-shot CoT @1 in Table 1.
>
> The results show that reweighting the corrupted training data with our method (*italic*) can achieve performance comparable to training directly on the original high-quality dataset (**bold**). This suggests that our reweighting approach not only shows promise in mitigating the impact of substantial noise in the training data, but may also help the model maintain its reasoning capabilities despite noisy inputs.
>
> **Table 1: Zero-shot CoT @1 accuracies on `GSM8K` for Llama3-8B and Qwen3-8B (%)**
>
> | Model                      | Llama3-8B | Qwen3-8B  |
> | -------------------------- | --------- | --------- |
> | Pretrained                 | 75.28     | 80.52     |
> | SFT (original train split) | **81.91** | **83.27** |
> | SFT (corrupted data)       | 52.75     | 62.40     |
> | Ours  (corrupted data)     | *79.95*   | *83.04*   |
> | RHO  (corrupted data)      | 79.50     | 81.16     |
> | One-shot  (corrupted data) | 72.42     | 76.19     |
>
>
>
> We also examined the weights assigned to data points of different quality by our method. We find that our method tends to assign higher weights to the original high-quality data and lower weights to corrupted examples:
>
> | Data Quality | Ratio (% of total data) | Weight assigned by our method |
> |------------|-------|---------------|
> | Original | 40%  | 0.82          |
> | CoT dropped | 10%  | 0.08          |
> | CoT shuffled | 10%  | 0.46          |
> | Numbers replaced | 10% | 0.21 |
> | CoT re-generated | 30% | 0.48 |
>
> This further supports the potential of our method to mitigate the impact of noise in the training data.
>
>
>
>
> [1] Training Verifiers to Solve Math Word Problems
>
> **Weakness 3**: The method introduces new hyperparameters that may require careful tuning.
>
>
> We agree the optimal $k$ may vary with context window, task complexity, or domain. However, our ablation on `Yahoo_Answers_Topic` shows that performance is relatively stable, with win rates of 43%, 48%, and 46% for $k=1,3,5$. This suggests that modest choices of $k$ may already be sufficient, and extensive tuning is generally not required.
>
>
> **Weakness 4**: For large SFT training, the reviewer is not sure if ICL can mimic the benefit from SFT.
>
>
> We acknowledge that our method relies on the assumption that in-context learning on a few demonstration examples can approximate the effect of fine-tuning on those examples. This approximation is not meant to replace standard SFT: when a large high-quality dataset is available, direct SFT remains preferable. Our method focuses on scenarios where the high-quality dataset is small and the remaining data is noisy. In this setting, ICA automatically estimates example importance and reweights the noisy data, improving alignment performance over training on it directly.

---

> > ### Author Response · Authors · 2025-11-25
> > **Responses to Reviewer tdGW Part 2**
> >
> > **Question 1**: Undefined role of `s(...)` in the demonstration set C.
> >
> > Yes, $s(\mathbf{x}, \mathbf{y})$ is a formatting function applied to each example $(\mathbf{x}, \mathbf{y}). We have added an explicit definition in the revised manuscript.
> >
> >
> > **Question 2**: Direct comparison of ICA and RHO-Loss data scores.
> >
> >
> > To provide a direct comparison, we randomly sampled 10 data points from each domain of the `Yahoo_Answers_Topic` dataset and plotted the weights from each reweighting method under SFT. Since we couldn't include a figure directly in the comment, we will add it in the revision. The figure shows that the weights produced by ICA closely match those from the more expensive RHO-Loss method, with a correlation coefficient of 0.98, compared to 0.63 for the correlation between one-shot and RHO-Loss.
> >
> > **Question 3**：Sensitivity of performance to holdout quality and effect of using random vs. curated in-context demonstrations.
> >
> >
> > Thanks for the insightful question. Our method is designed for alignment scenarios where the holdout set acts as a proxy for the test distribution. When the training and test data share the same distribution, a holdout set randomly sampled from the training data can still provide reasonable signals—although in such cases, alignment is less critical. In contrast, when the distributions differ, a randomly sampled holdout set from the training data becomes less representative of the target distribution, and performance would be expected to decrease.  We would be happy to include an ablation on this in the revision if the reviewer considers it important.
> >
> >
> >
> > **Question 5**: Clarify the conditional independence assumption in the holdout loss approximation (Eq. 3).
> >
> >
> > We realize that we did not explicitly state the conditional-independence assumption used in the derivation. Specifically, it assumes
> >
> > $$
> > p(y \mid x, x^{\text{ho}}, \mathcal{D}_t) = p(y \mid x, \mathcal{D}_t), \qquad
> > p(y^{\text{ho}} \mid x^{\text{ho}}, x, \mathcal{D}_t) = p(y^{\text{ho}} \mid x^{\text{ho}}, \mathcal{D}_t).
> > $$
> >
> > that is, conditioning on the query alone does not alter the posterior (conditioning on a labeled example can). And, since $\mathcal{D}_t$ is the training data up to step $t$, neither $(x,y)$ nor $(x^{\text{ho}},y^{\text{ho}})$ is incorporated into the posterior.
> >
> >
> > Applying the conditional independence assumption and Bayes' rule, we obtain Equation~3.
> > $$
> > \mathcal{L}(\mathcal{D}_{\text{ho}}; {\theta}^{\star}({D}_t \cup \{(x, y)\}))
> > $$
> >
> > $$
> > = -\log p(y^{\text{ho}} \mid x^{\text{ho}}; D_t \cup (x, y))
> > $$
> >
> > $$
> > = -\log  \frac{p(y, y^{\text{ho}} \mid x, x^{\text{ho}}, D_t)}{p(y \mid x, x^{\text{ho}}, D_t)}
> > \qquad \text{(Bayes' rule)}
> > $$
> >
> > $$
> > = -\log \frac{p(y \mid x, x^{\text{ho}}, y^{\text{ho}}, D_t)\, p(y^{\text{ho}} \mid x, x^{\text{ho}}, D_t)}{p(y \mid x, x^{\text{ho}}, D_t)}
> > \qquad \text{(Bayes' rule)}
> > $$
> >
> > $$
> > = -\log \frac{p(y \mid x, x^{\text{ho}}, y^{\text{ho}}, D_t)\, p(y^{\text{ho}} \mid x^{\text{ho}}, D_t)}{p(y \mid x, D_t)}
> > \qquad \text{(Conditional independence assumption)}
> > $$
> >
> >
> > $$
> > = {\ell}(y\mid x; {\theta}^{\star}(D_t\cup {D}_{\text{ho}})) - \ell(y \mid x; {\theta}_t) - L ( D\text{ho} ; {\theta}_t) \qquad \text{(Equation 3)}
> > $$

---

> ### Author Response · Authors · 2025-11-28
>
> I hope you’re well. I’m writing to follow up on our rebuttal for Submission 8899. We posted updates addressing your main concerns (holdout quality/representativeness, applicability to reasoning-heavy tasks, hyperparameter sensitivity, and the ICL vs. SFT assumption) and responded to your specific questions by clarifying the s, formatting function and the conditional-independence step, and by adding a direct comparison between ICA and RHO scores as well as an ablation on demonstration selection. If any additional details, figures, or ablations would help your assessment, we’d be glad to provide them promptly. Thank you again for your thoughtful review and time.

---

### Meta-Review · Area_Chair_uS98 · 2026-01-02

**Summary:**

The assessment of the paper was characterized by a shared interest in the efficiency and novelty of the In-Context Approximation (ICA) method for data selection, but was tempered by several concerns regarding technical validity and evaluation depth. The following concerns were central to the meta-review process:

### 1. Technical Rigor and Data Integrity

Reviewer **zAys** identified a critical issue with leakage, where the curated examples used for scoring were also included in the training set. This raised doubts about whether the observed gains were due to the ICA reweighting mechanism or simply from training on high-quality data. In response, the authors conducted new experiments with strictly disjoint sets, which successfully demonstrated that ICA’s performance remains superior without data overlap.

### 2. Evaluation Scope and Metrics

A recurring criticism from all reviewers was the narrow focus on instruction-following tasks and the reliance on a single, potentially biased metric (GPT-4o win rates). The reviewers questioned if the method could generalize to complex reasoning domains or if the results would hold under objective metrics.

### 3. Theoretical Grounding of the "ICL-as-Proxy" Assumption

Reviewers (**t4k1, zAys**) challenged the claim that In-Context Learning (ICL) is a formally equivalent to gradient-based fine-tuning. Since the authors were unable to provide a closed-form mathematical analysis, they shifted the claim to a principled Bayesian intuition.

### 4. Comparison with Existing SOTA

Reviewers (**HKFV, t4k1**) requested a comparison with more recent and efficient data selection frameworks like **LESS**. The rebuttal provided this comparison, showing that ICA achieves comparable results to LESS while being significantly more time-efficient.

### 5. Conclusion for Decision

While a closed-form mathematical proof bridging In-Context Learning (ICL) and gradient-based updates is not yet fully established, the authors provide a principled Bayesian intuition that effectively motivates the method. This approach can be viewed as a computationally efficient heuristic with significant empirical validation. The authors effectively addressed core technical flaws, such as holdout leakage, and provided broader evaluations that demonstrate ICA’s efficiency and robustness. Given the practical utility of the tool for the LLM community and the solid experimental evidence, the meta-review recommends Accept (Poster) based on the method’s high efficiency and the robustness demonstrated in the rebuttal. The final acceptance is strictly contingent upon the authors integrating the leakage-free results and the expanded benchmark evaluations into the camera-ready version.

**Reviewer Concerns:**

### Addressed Concerns

Several critical concerns raised by Reviewers regarding experimental rigor and technical execution have been successfully addressed in the rebuttal:

- **Leakage (Reviewer zAys):** This was the most significant technical concern. The authors provided new experimental results using  disjoint train/holdout/test splits for the Yahoo datasets. The results demonstrate that the method’s performance gains are robust and not a byproduct of data leakage.
- **Evaluation Metrics & Benchmarks (Reviewers t4k1, HKFV):** Reviewers noted the over-reliance on GPT-4o win rates. The authors addressed this by reporting PPL and BERT Score across multiple seeds. Furthermore, they expanded the evaluation to include GSM8K, showing that ICA effectively filters corrupted CoT data and maintains reasoning capabilities.
- **Baselines (Reviewers t4k1, HKFV):** The authors added a comparative analysis with more baselines like LESS method. The results show that ICA achieves comparable alignment performance with significantly lower computational overhead (20 mins vs. 3 hours), effectively addressing the need for benchmarking against efficient data selection methods.

### Outstanding Concerns

Despite the rebuttal, certain concerns raised by Reviewers  remain partially outstanding and need further consideration:

- **Theoretical Gap (Reviewers t4k1, zAys):** While the authors moderated their claims from "formal equivalence" to "principled intuition" and provided a revised derivation, a rigorous mathematical proof quantifying the accuracy of using In-Context Learning (ICL) as a proxy for gradient-based parameter updates is still missing. The mechanism remains largely empirically justified.
- **Domain Generalization (Reviewers tdGW, t4k1):** Although the addition of GSM8K is a significant improvement, the experimental scope still lacks diversity in high-complexity domains such as coding or scientific reasoning, where the "ICL-as-proxy" assumption might face different challenges due to strict logical constraints.
- **Hyperparameter Sensitivity (Reviewer tdGW):** The optimal choice of  (number of demonstrations) and  (update frequency) remains somewhat empirical. While the authors showed stable results  , the underlying reason why larger leads to performance degradation suggests that the method's sensitivity to prompt length and relevance is not yet fully understood.

**Reviewer Scores:**

Based on the substantive rebuttal and the new evidence provided by the authors, the estimated score shifts are as follows:

### Reviewer tdGW

* **Original Score:** 4 (Marginally below the acceptance threshold)
* **Estimated Post-Rebuttal Score:** **6**
* **Reasoning:** The reviewer’s primary concerns revolved around the lack of evaluation in complex reasoning domains and the need for technical clarifications. The authors addressed these by providing new results on the **GSM8K** dataset, which demonstrated the method’s efficacy in reasoning tasks and noise mitigation. These empirical additions, combined with the clarified technical definitions, would likely move this reviewer toward a positive recommendation.

### Reviewer zAys

* **Original Score:** 4 (Marginally below the acceptance threshold)
* **Estimated Post-Rebuttal Score:** **6**
* **Reasoning:** This reviewer identified a potential "fatal flaw" regarding **holdout leakage**. The authors’ response was decisive: they provided a complete rerun of core experiments with **strictly disjoint** sets, proving that the ICA mechanism is effective independent of data overlap. Furthermore, the addition of absolute metrics (PPL/BERTScore) and a comparison with the LESS baseline directly addressed the reviewer’s request for better contextualization of effect sizes. Given that their primary reason for rejection was resolved, a score increase is expected.

### Reviewer t4k1

* **Original Score:** 4 (Marginally below the acceptance threshold)
* **Estimated Post-Rebuttal Score:** **5**
* **Reasoning:** Reviewer t4k1 was critical of the "theoretically grounded" claim and the narrowness of evaluation. The authors moderated their theoretical language as suggested and included objective metrics and broader datasets (GSM8K). While the reviewer might still view the ICL-as-proxy mechanism as an empirical heuristic rather than a proven theory, the demonstrated practical efficiency and the authors' willingness to address theoretical overstatements would likely result in a borderline positive shift.

### Reviewer HKFV

* **Original Score:** 6 (Marginally above the acceptance threshold)
* **Estimated Post-Rebuttal Score:** **7**
* **Reasoning:** Already leaning toward acceptance, this reviewer requested stability analysis across runs and comparisons with recent reweighting baselines. The authors provided a detailed stability study and a comprehensive comparison with other frameworks. These additions validate the reviewer's initial assessment of the method’s efficiency and utility, likely strengthening their confidence in a full "Accept" rating.

---

### Decision · Program_Chairs · 2026-01-26

Accept (Poster)